# RETRIEVER: LEARNING CONTENT-STYLE REPRESENTATION AS A TOKEN-LEVEL BIPARTITE GRAPH

**Dacheng Yin**[1]*,**Xuanchi Ren**[2]*,**Chong Luo**[3]†,**Yuwang Wang**[3]†,**Zhiwei Xiong**[1] ,**Wenjun Zeng**[4]
[1]University of Science and Technology of China, [2]HKUST, [3]Microsoft Research Asia, [4]EIT
[1]{ydc@mail., zwxiong@}ustc.edu.cn,  [2]xrenaa@connect.ust.hk,
[3]{cluo, yuwwan}@microsoft.com,    [4]wenjunzeng@eias.ac.cn

## ABSTRACT

This paper addresses the unsupervised learning of content-style decomposed representation. We first give a definition of style and then model the content-style representation as a token-level bipartite graph. An unsupervised framework, named `Retriever`, is proposed to learn such representations. First, a cross-attention module is employed to retrieve permutation invariant (P.I.) information, defined as style, from the input data. Second, a vector quantization (VQ) module is used, together with man-induced constraints, to produce interpretable content tokens. Last, an innovative link attention module serves as the decoder to reconstruct data from the decomposed content and style, with the help of the linking keys. Being modal-agnostic, the proposed `Retriever` is evaluated in both speech and image domains. The state-of-the-art zero-shot voice conversion performance confirms the disentangling ability of our framework. Top performance is also achieved in the part discovery task for images, verifying the interpretability of our representation. In addition, the vivid part-based style transfer quality demonstrates the potential of `Retriever` to support various fascinating generative tasks. Project page at `https://ydcustc.github.io/retriever-demo/`.

## 1 INTRODUCTION

Human perceptual systems routinely separate content and style to better understand their observations (Tenenbaum & Freeman, 2000). In artificial intelligence, a content and style decomposed representation is also very much desired. However, we notice that existing work does not have a unified definition for content and style. Some definitions are dataset-dependent (Chou & Lee, 2019; Ren et al., 2021), while some others have to be defined on a certain modality (Lorenz et al., 2019; Wu et al., 2019). We wonder, since content-style separation is helpful to our entire perception system, why is there not a unified definition that applies to all perception data?

In order to answer this question, we must first study the characteristics of data. The data of interest, including text, speech, image, and video, are structured. They can be divided into standardized tokens, either naturally as words in language and speech or intentionally as patches in images. Notably, the order of these tokens matters. Disrupting the order of words can make a speech express a completely different meaning. Reversing the order of frames in a video can make a stand-up action become a sit-down action. But there is also some information that is not affected by the order of tokens. For example, scrambling the words in a speech does not change the speaker's voice, and a frame-shuffled video does not change how the person in the video looks. We notice that the content we intuitively think of is affected by the order of tokens, while the style is usually not. Therefore, we could generally define style as token-level permutation invariant (P.I.) information and define content as the rest of the information in structured data.

However, merely dividing data into two parts is not enough. As Bengio et al. (2013) pointed out, if we are to take the notion of disentangling seriously, we require a richer interaction of features than that offered by simple linear combinations. An intuitive example is that we could never generate

---

*Equal contribution. Work done during internships at Microsoft Research Asia.
†Joint corresponding author.

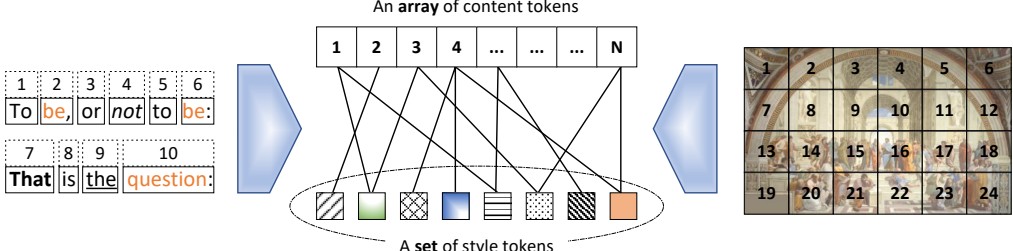

Figure 1: Illustration of the proposed modal-agnostic bipartite-graph representation of content and style. The content is an array of position-sensitive tokens and the style is a set of P.I. tokens.

a colored pattern by linearly combining a generic color feature with a gray-scale stimulus pattern. Inspired by this, we propose to model content and style by a token-level bipartite graph, as Figure 1 illustrates. This representation includes an array of content tokens, a set of style tokens, and a set of links modeling the interaction between them. Such a representation allows for fine-grained access and manipulation of features, and enables exciting downstream tasks such as part-level style transfer.

In this paper, we design a modal-agnostic framework, named `Retriever`, for learning bipartite graph representation of content and style. `Retriever` adopts the autoencoder architecture and addresses two main challenges in it. One is how to decompose the defined content and style in an unsupervised setting. The other is how to compose these two separated factors to reconstruct data.

To tackle the first challenge, we employ a cross-attention module that takes the dataset-shared prototype as query to retrieve the style tokens from input data (Carion et al., 2020). A cross-attention operation only allows the P.I. information to pass (Lee et al., 2019), which is exactly what we want for style. On the other content path, we employ a vector quantization (VQ) module (van den Oord et al., 2017) as the information bottleneck. In addition, we enforce man-induced constraints to make the content tokens interpretable. To tackle the second challenge, we innovate the link attention module for the reconstruction from the bipartite graph. Specifically, the content and style serve as the query and value, respectively. Links between the content and style are learnt and stored in the linking keys. Link attention allows us to retrieve style by content query. As such, the interpretability is propagated from content to style, and the entire representation is friendly to fine-grained editing.

We evaluate `Retriever` in both speech and image domains. In the speech domain, we achieve state-of-the-art (SOTA) performance in zero-shot voice conversion, demonstrating a complete and precise decomposition of content and style. In the image domain, we achieve competitive results in part discovery task, which demonstrates the interpretability of the decomposed content. More excitingly, we try part-level style transfer, which cannot be offered by most of the existing content-style disentanglement approaches. Vivid and interpretable results are achieved.

To summarize, our main contributions are three-folds: i) We provide an intuitive and modal-agnostic definition of content and style for structured data. We are the first to model content and style with a token-level bipartite graph. ii) We propose an unsupervised framework, named `Retriever`, for learning the proposed content-style representation. A novel link attention module is designed for data reconstruction from content-style bipartite graph. iii) We demonstrate the power of `Retriever` in challenging downstream tasks in both speech and image domains.

## 2 RELATED WORK

Content-style decomposed representation can be approached in supervised or unsupervised settings. When style labels, such as the speaker labels of speeches (Kameoka et al., 2018; Qian et al., 2019; Yuan et al., 2021) and the identity labels of face images (Mathieu et al., 2016; Szabó et al., 2018; Jha et al., 2018; Bouchacourt et al., 2018; Gabbay & Hoshen, 2020), are available, latent variables can be divided into content and style based on group supervision.

Recently, there has been increased interest in unsupervised learning of content and style. Since there is no explicit supervision signal, the basic problem one must first solve is the definition of content and style. We discover that all existing definitions are either domain-specific or task-specific. For

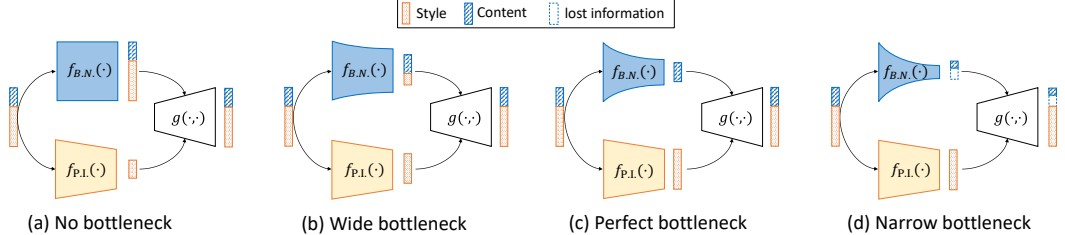

| Style | Content | lost information |

(a) No bottleneck     (b) Wide bottleneck     (c) Perfect bottleneck     (d) Narrow bottleneck

Figure 2: An illustration of our content-style separation mechanism.

example, in speech domain, Chou & Lee (2019) assume that style is the global statistical information and content is what is left after instance normalization (IN). Ebbers et al. (2021) suggest that style captures long-term stability and content captures short-term variations. In image domain, the definition is even more diverse. Lorenz et al. (2019) try to discover the invariants under spatial and appearance transformations and treat them as style and content, respectively. Wu et al. (2019) define content as 2D landmarks and style as the rest of information. Ren et al. (2021) define content as the most important factor across the whole dataset for image reconstruction, which is rather abstract. In this work, we attempt to find a general and modal-agnostic definition of content and style.

Style transfer is partially related to our work, as it concerns the combination of content and style to reconstruct data. AdaIN (Huang & Belongie, 2017; Chou & Lee, 2019) goes beyond the linear combination of content and style (Tenenbaum & Freeman, 2000) and proposes to inject style into content by aligning the mean and variance of the content features with those of the style features. However, style is not separated from content in this line of research. Liu et al. (2021) touch upon the part-based style transfer task as we do. They model the relationship between content and style by a one-one mapping. They follow the common definition of content and style in the image domain as shape and appearance, and try to disentangle them with hand-crafted data augmentation methods.

Besides, the term "style" is often seen in image generative models, such as StyleGAN (Karras et al., 2019). However, the style mentioned in this type of work is conceptually different from the style in our work. In StyleGAN, there is no concept of content, and style is the whole latent variable containing all the information including the appearance and shape of an image. Following StyleGAN, Hudson & Zitnick (2021) employ a bipartite structure to enable long-range interactions across the image, which iteratively propagates information from a set of latent variables to the evolving visual features. Recently, researchers have become interested in disentangling content and style from the latent variables of StyleGAN (Alharbi & Wonka, 2020; Kwon & Ye, 2021). However, they only work for well-aligned images and are hard to be applied to other modalities.

## 3 CONTENT-STYLE REPRESENTATION FOR STRUCTURED DATA

In this section, we provide definitions of content and style for structured data, introduce the framework for content-style decomposition, and propose the token-level bipartite graph representation.

### 3.1 DEFINITION OF CONTENT AND STYLE

The data of interest is structured data that can be tokenized, denoted by $\boldsymbol{X} = [\boldsymbol{x}_1, \boldsymbol{x}_2, ..., \boldsymbol{x}_n]$. We think of text, speech, and image, among many others, as structured data. Each token $\boldsymbol{x}_i$ can be a word in text, a phoneme in a speech, or a patch in an image. These data are structured because non-trivial instances (examples of trivial instances are a silent speech or a blank image) are not able to keep their full information when the order of tokens is not given. Inspired by this intuition, we define style of $\boldsymbol{X}$ as the information that is not affected by the permutation of tokens, or permutation invariant (P.I.) information. Content is the rest of information in $\boldsymbol{X}$.

### 3.2 CONTENT-STYLE SEPARATION

The information in a piece of structured data is carried either in the content or in the style. By definition, style can be extracted by a P.I. function $f_{P.I.}(\cdot)$, which satisfies $f(\pi(\boldsymbol{X})) = f(\boldsymbol{X})$, where $\pi(\cdot)$ represents permutation of tokens. To achieve content-style decomposition, we naturally

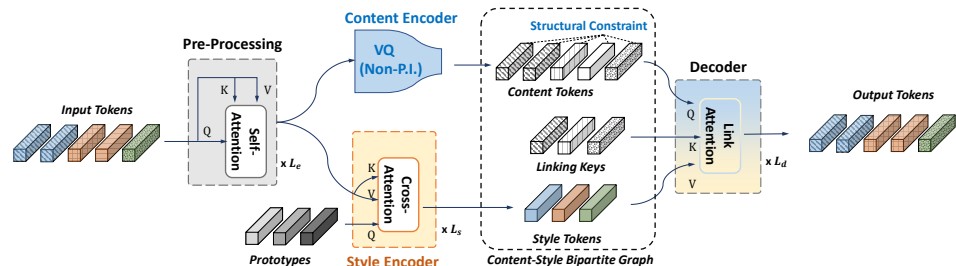

Figure 3: Overview of the proposed `Retriever` framework. The name is dubbed because of the dual-retrieval operations: the cross-attention module retrieves style for content-style separation, and the link attention module retrieves content-specific style for data reconstruction.

adopt an autoencoder architecture, as shown in Figure 2. The bottom path is the style path which implements $f_{P.I.}(\cdot)$. We shall find a powerful P.I. function which will let all the P.I. information pass. The top path is responsible for extracting the content, but the challenge is that there does not exist a function which only lets pass the non-P.I. information. Therefore, we employ in the content path a permutation-variant function, which will let all information pass, including the P.I. information. Obviously, if we do not pose any constraint on the content path, as shown in Figure 2 (a), the style information will be leaked to the content path. To squeeze the style information out of the content path, an information bottleneck $f_{B.N.}(\cdot)$ is required. A perfect bottleneck, as shown in Figure 2 (c), can avoid the style leak while achieving perfect reconstruction. But an imperfect bottleneck, being too wide or too narrow, will cause style leak or content loss, as shown in Figure 2 (b) and Figure 2 (d), respectively.

### 3.3 TOKEN-LEVEL BIPARTITE GRAPH REPRESENTATION OF CONTENT AND STYLE

After dividing the structured input $X$ into content feature $C$ and style feature $S$, we continue to explore how the relationship between $C$ and $S$ should be modeled. Ideally, we do not want an all-to-one mapping, in which the style is applied to all the content as a whole. Nor do we want a one-to-one mapping where each content token is associated with a fixed style. In order to provide a flexible way of interaction, we propose a novel token-level bipartite graph modeling of $C$ and $S$.

The token-level representations of $C$ and $S$ are $C = [c_1, c_2, ..., c_n]$ and $S = \{s_1, s_2, ..., s_m\}$, respectively. Note that there is a one-one correspondence between $x_i$ and $c_i$, so the order of the structured data is preserved in the content $C$. While the order is preserved, the semantic meaning of each $c_i$ is not fixed. This suggests that the bipartite graph between $C$ and $S$ cannot be static. In order to model a dynamic bipartite graph, we introduce a set of learnable linking keys $K = \{k_1, k_2, ..., k_m\}$. The linking keys and style tokens form a set of key-value pairs $\{(k_i, s_i)\}_{i=1}^{m}$. Our linking key design allows for a soft and learnable combination of content and style. The connection weight between a content token $c_i$ and a style token $s_j$ is calculated by $w_{i,j} = r_\theta(c_i, k_j)$, where $r_\theta$ is a learnable linking function parameterized with $\theta$. For a content token $c_i$, its content-specific style feature now can be calculated by $s_\theta(c_i) = \sum_{j=1}^{m} \hat{w}_{i,j} s_j$, where $\hat{w}_{i,j}$ are normalized weights.

## 4 THE RETRIEVER FRAMEWORK

### 4.1 OVERVIEW

This section presents the design and implementation of the `Retriever` framework, which realizes the bipartite graph representation of content and style as described in Section 3. `Retriever`, as shown in Figure 3, is an autoencoder comprising of four major blocks, two of which implement the most important style retrieval functions. This is why the name `Retriever` is coined.

`Retriever` is modal-agnostic. The input of the framework is an array of input tokens, denoted by $X = [x_1, x_2, .., x_n] \in \mathbb{R}^{n \times d}$, obtained by modal-specific tokenization operations from raw data. The first major component is a pre-processing module to extract the feature $F \in \mathbb{R}^{n \times d}$ from $X$. We implement it as a stack ($L_e$) of non-P.I. transformer encoder blocks (Vaswani et al., 2017) to keep all the important information. Then $F$ is decomposed by the content encoder $f_{B.N.}(\cdot)$

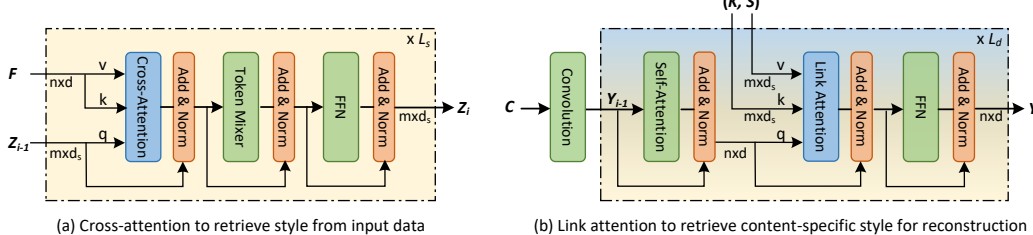

(a) Cross-attention to retrieve style from input data      (b) Link attention to retrieve content-specific style for reconstruction

Figure 4: Implementation of (a) the style encoder and (b) the decoder in `Retriever`.

and the style encoder $f_{P.I.}(\cdot)$. $f_{B.N.}(\boldsymbol{X})$ is implemented by vector quantization to extract content $\boldsymbol{C} = [\boldsymbol{c}_1, \boldsymbol{c}_2, ..., \boldsymbol{c}_n]$. Structural constraint can be imposed on $\boldsymbol{C}$ to improve interpretability. A cross-attention module is employed as $f_{P.I.}(\boldsymbol{X})$ to retrieve style $\boldsymbol{S} = \{\boldsymbol{s}_1, \boldsymbol{s}_2, ..., \boldsymbol{s}_m\}$. At last, the decoder, implemented by the novel link attention module, reconstructs $X$ from the bipartite graph.

## 4.2 CONTENT ENCODER

Content encoder should serve two purposes. First, it should implement an information bottleneck to squeeze out the style information. Second, it should provide interpretability such that meaningful feature editing can be enabled.

**Information bottleneck.** Vector quantization (VQ) is a good candidate of information bottleneck (Wu et al., 2020), though many other choices also easily fit into our framework. VQ maps data from a continuous space into a set of discrete representations with restricted information capacity, and has shown strong generalization ability among data formats, such as image (van den Oord et al., 2017; Razavi et al., 2019; Esser et al., 2021), audio (Baevski et al., 2020a;b) and video (Yan et al., 2021).

We use product quantization (Baevski et al., 2020a;b), which represents the input $\boldsymbol{F}$ with a concatenation of $G$ groups of codes, and each group of codes is from an independent codebook of $V$ entries. To encourage all the VQ codes to be equally used, we use a batch-level VQ perplexity loss, denoted as $\mathcal{L}_{VQ}$. Please refer to Appendix A for more details. When the bottleneck capacity is appropriate, the content can be extracted as $\boldsymbol{C} = \text{VQ}(\boldsymbol{F})$, where $\boldsymbol{C} \in \mathbb{R}^{n \times d_c}$.

**Interpretability.** Priori knowledge can be imposed on the content path of `Retriever` to increase the interpretability of feature representations. In this work, we introduce structural constraint $\mathcal{L}_{SC}$ to demonstrate this capability. Other prior knowledge, including modal-specific ones, can be added to the framework as long as they can be translated into differentiable loss functions.

The structural constraint we have implemented in `Retriever` is a quite general one, as it reflects the locality bias widely exists in natural signals. In image domain, we force the spatially adjacent tokens to share the same VQ code, so that a single VQ code may represent a meaningful object part. In speech domain, we discourage the VQ code from changing too frequently along the temporal axis. As such, adjacent speech tokens can share the same code that may represent a phoneme. The visualization in the next section will demonstrate the interpretability of `Retriever` features.

## 4.3 STYLE ENCODER WITH CROSS-ATTENTION

We have defined style as the P.I. information that can be extracted by a P.I. function from the structured data. Cross-Attention, which is widely used in set prediction task (Carion et al., 2020) and set data processing such as 3D point cloud (Lee et al., 2019), is known to be a P.I. function. It is also a powerful operation that can project a large input to a smaller or arbitrary target shape. Previously, Perceiver (Jaegle et al., 2021b) and Perceiver IO (Jaegle et al., 2021a) have used this operation as a replacement of the self-attention operation in Transformers to reduce the complexity. However, cross-attention operation does not preserve data structure. In order to compensate for this, additional positional embedding is implemented to associate the position information with each input element.

Conversely, our work takes advantage of cross-attention's non-structure-preserving nature to retrieve the P.I. style information. A key implementation detail is that position information should not be associated with the input tokens. Otherwise, there will be content leaks into the style path. To implement the style encoder, we follow previous work (Carion et al., 2020; Lee et al., 2019) to learn

$m$ dataset-shared prototypes $\boldsymbol{Z}_0 \in \mathbb{R}^{m \times d_s}$ as seed vectors. Figure 4 (a) shows the implementation details of the style encoder, which is a stack of $L_s$ cross-attention blocks, token mixing layers, and feed-forward network (FFN) layers. The final output is the retrieved style $\boldsymbol{S} \in \mathbb{R}^{m \times d_s}$.

## 4.4 DECODER WITH LINK ATTENTION

In an autoencoder architecture, the decoder is responsible for reconstructing data from latent features. In learning a content-style representation, the ability of the decoder limits the way content and style can be modeled. Therefore, decoder design becomes the key to the entire framework design.

In order to support the bipartite graph representation of content and style, we innovatively design link attention by unleashing the full potential of multi-head attention. Unlike self-attention and cross-attention, link attention has distinct input for $Q$, $K$, and $V$. In the `Retriever` decoder, content tokens are queries and style tokens are values. The keys are a set of learnable linking keys $\boldsymbol{K} \in \mathbb{R}^{m \times d_s}$ which are paired with the style tokens to represent the links in the content-style bipartite graph. Such a design allows us to retrieve content-specific style for data reconstruction, enabling fine-grained feature editing and interpretable style transfer.

Figure 4 (b) shows the implementation details of the `Retriever` decoder. We first use a convolutional layer directly after VQ to aggregate the contextual information as $\boldsymbol{Y}_0 = \text{Conv}(\boldsymbol{C})$. Then the style is injected into content for $L_d$ times. In each round of style injection, we add a self-attention layer before the link attention layer and append an FFN layer after it. The element order in $\boldsymbol{K}$ and $\boldsymbol{S}$ does not matter, as long as $\boldsymbol{k}_i$ and $\boldsymbol{s}_i$ are paired. The output of the decoder is then detokenized to reconstruct the raw data, whose difference with the input raw data is used as the reconstruction loss.

## 4.5 LOSS FUNCTIONS

Our final loss function $\mathcal{L}_{sum}$ consists of three components: the reconstruction loss $\mathcal{L}_{rec}$, the VQ diversity loss $\mathcal{L}_{VQ}$, and the structural constraint loss $\mathcal{L}_{SC}$ (see implementation details in Appendix D):

$$\mathcal{L}_{sum} = \lambda_{rec}\mathcal{L}_{rec} + \lambda_{VQ}\mathcal{L}_{VQ} + \lambda_{SC}\mathcal{L}_{SC}, \tag{1}$$

where $\lambda_{rec}, \lambda_{VQ}$, and $\lambda_{SC}$ are hyper-parameters controlling the weights of the three losses.

## 5 EXPERIMENTS

We evaluate the `Retriever` framework in both speech and image domains. Due to the differences in tasks, datasets, and evaluation metrics, we organize the experiments in these two domains in two subsections. We use visualizations or quantitative results to: i) demonstrate the effect of content-style separation and the interpretability of content; ii) illustrate how the bipartite graph representation conveys the interpretability of content to style and supports fine-grained style transfer.

### 5.1 SPEECH DOMAIN

#### 5.1.1 TRAINING RETRIEVER FOR SPEECH SIGNALS

`Retriever` for speech signals is trained with the entire 44-hour CSTR VCTK Corpus (Veaux et al., 2017) containing 109 speakers. We use the pre-trained CPC network (Rivière et al., 2020) followed by a depth-wise convolution to perform tokenization. In `Retriever`, the content encoder is implemented by a VQ with two groups ($G = 2$) and $V = 100$. A truncated neighborhood cross-entropy loss is used as $\mathcal{L}_{SC}$ to enforce the structural constraint, which is only applied to group #0. For the style encoder, we set the number of style tokens to 60. At the decoder, the content is first processed by a 1D depth-wise convolution with kernel size 31 before being passed to link attention. In the detokenization step, we resample the output log-Mel spectrum to 80Hz and feed it into Parallel WaveGAN vocoder (Yamamoto et al., 2020) to generate the waveform. We apply L1 reconstruction loss on log-Mel spectrum as $\mathcal{L}_{rec}$. More details can be found in Appendix F.1.

#### 5.1.2 VISUALIZATION OF CONTENT-STYLE REPRESENTATION

To understand how `Retriever` models the content, the style, and the bipartite graph between them, we visualize the VQ codes and the decoder link attention map for a 2.5s-long test utterance in

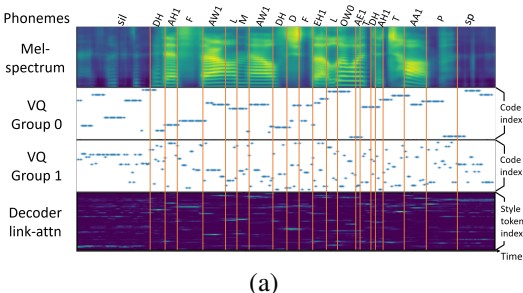 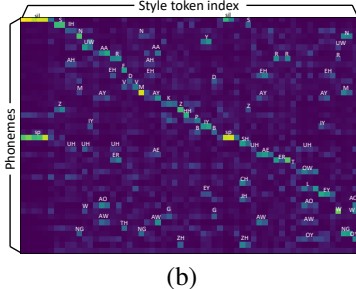

| (a) | (b) |

Figure 5: Visualization of (a) VQ codes, decoder link attention map, and the corresponding Mel spectrum; (b) co-occurrence map between ground-truth phoneme and style tokens. Strong co-occurrence is labeled with phoneme names. For higher resolution see Appendix B.

Table 1: Phoneme information probing.

| Settings | Frame | Frame w/ context |
|---|---|---|
| CPC + k-means | 9.6% | 9.5% |
| Group #0 | 48.1% | 62.7% |
| Group #1 | 21.0% | 45.9% |
| Group #0, #1 | 52.3% | 66.9% |

Table 2: Comparison with other methods.

| Methods | SV Accu | MOS | SMOS |
|---|---|---|---|
| AutoVC (Qian et al., 2019) | 17.9% | 1.65±0.11 | 1.92±0.13 |
| AdaIN-VC (Chou & Lee, 2019) | 46.5% | 1.86±0.10 | 2.26±0.14 |
| FragmentVC (Lin et al., 2021b) | 89.5% | 3.43±0.12 | 3.54±0.15 |
| S2VC (Lin et al., 2021a) | 96.8% | 3.18±0.12 | 3.36±0.15 |
| Retriever | **99.4%** | **3.44±0.13** | **3.84±0.14** |

Figure 5 (a). Each dot represents one token and the token hop is 10ms. Retriever encodes content with VQ, and we further impose structural constraint to VQ Group #0. We find that this group of code demonstrates good interpretability as desired. Codes for adjacent tokens are highly consistent, and the changes of codes are well aligned with ground-truth phoneme changes. The VQ codes of group #1 exhibit a different pattern. They switch more frequently as they capture more detailed residual variations within each phoneme. We further study the interpretability of content by phoneme information probing (implementation described in Appendix F.3), which is used to show the information captured by the discovered content units (Chorowski et al., 2019). We measure the frame-level phoneme classification accuracy with or without context and compare with the baseline (CPC + k-means). Results in Table 1 show that the two groups of VQ codes discovered in our unsupervised framework align surprisingly well with the annotated phoneme labels. Notably, group #0 codes capture most of the phoneme information and group #1 codes are less interpretable. This demonstrates how the structural constraint helps to increase the interpretability of content.

In decoder link attention, all the content tokens within the same phoneme tend to attend to the same style vector, while tokens from different phonemes always attend to different style vectors. We further visualize the co-occurrence map between the ground-truth phonemes and the style vectors in Figure 5 (b) (calculation described in Appendix B). We see that the phonetic content and style vectors have a strong correlation. Interestingly, each style vector encodes the style of a group of similarly pronounced phonemes, such as the group of 'M', 'N', and 'NG'. This observation confirms that content-specific style is actually realized in Retriever features.

### 5.1.3 ZERO-SHOT VOICE CONVERSION TASK

Zero-shot voice conversion task converts the source speaker's voice into any out-of-training-data speaker's while preserving the linguistic content. We follow the setting of previous work to randomly select 1,000 test utterance pairs of 18 unseen speakers from the CMU Arctic databases (Kominek & Black, 2004). Each conversion is given five target utterances to extract the style. Both objective and subjective metrics are used to evaluate conversion similarity (SV accuracy, SMOS) and speech quality (MOSNet score, MOS). More details of the four metrics are in Appendix F.2.

Existing approaches fall into two categories: content-style disentanglement approaches (Qian et al., 2019; Yuan et al., 2021; Chou & Lee, 2019; Wu et al., 2020), and deep concatenative approaches (Lin et al., 2021b;a). Methods in the first category use a shared style embedding for all the content features while methods in the second category try to find the matching style for each content frame, resulting in high computational complexity. Our method provides a similar level of style granularity as the second class of approaches. But our bipartite graph representation is more compact and effective as it is capable of retrieving content-specific style with a light-weight link attention operation. Table 2 shows the comparison between our Retriever-based method and four SOTA methods. The superior performance demonstrates the power of the content-style bipartite graph representation.

### 5.1.4 Ablation Study on Retriever

Based on the zero-shot voice conversion task, we carry out ablation studies on the VCTK + CMU Arctic dataset to better understand Retriever.

**Style Encoder.** Unlike most previous work, which are limited to learning a single speaker style vector, our method can learn an arbitrary number of style tokens. When the number of style tokens increases from 1 to 10 and 60, the SV accuracy increases from 81.3% to 94.3% and 99.4%, showing the benefits of fine-grained style modeling.

**Content Encoder.** We study how the capacity of the information bottleneck affects the separation of content and style. Compared to the proper bottleneck setting ($G = 2, V = 100$), speech quality significantly drops when the bottleneck is "too narrow" ($G = 1, V = 10$) and conversion similarity drops when the bottleneck is "too wide" ($G = 8, V = 100$), as shown in Table 3. This indicates content loss and style leakage when the bottleneck is too narrow and too wide, respectively.

Table 3: Voice conversion quality at other content encoder and decoder settings.

| Settings | SV Accu | MOSNet |
|---|---|---|
| AdaIN-decoder | 72.2% | 2.89 |
| Too narrow B.N. | **99.8%** | 2.96 |
| Too wide B.N. | 90.5% | 3.10 |
| Proper B.N. | **99.4%** | **3.12** |

**Decoder.** To illustrate the advantages of our link attention-based decoder, we replace it with the traditional AdaIN module for experiment. As the AdaIN module only takes a single vector as input, we flatten the 60 style tokens produced by Retriever into a very long vector. Results in Table 3 show that, although the amount of information provided to the decoder is the same, AdaIN-based decoder has a significant performance drop. This confirms the value of link attention.

## 5.2 Image Domain

The image domain is one of the first domains to carry out the research on content-style disentanglement. Conventionally, shape is treated as content and appearance is treated as style. Interestingly, our unified definition of content and style aligns with the intuitive understanding of images.

### 5.2.1 Training Retriever for Images

To tokenize the image, we downsample it by 4 times using a stack of convolutions. For VQ module, we set $G$ to 1, and $V$ is dependent on the dataset. The convolution after VQ operation is of kernel size $3 \times 3$. We detokenize the output tokens to images by convolutions and PixelShuffles (Shi et al., 2016). For the reconstruction loss $\mathcal{L}_{rec}$, we apply a typical perceptual loss (Chen & Koltun, 2017). For the structural constraint $\mathcal{L}_{SC}$, we adopt a geometric concentration loss (Hung et al., 2019). We choose two commonly used datasets: **Celeba-Wild** (Liu et al., 2015) and **DeepFashion** (Liu et al., 2016). Images in Celeba-Wild are not aligned and each face has five landmark coordinates. The full-body images from DeepFashion are used. See Appendix E for more details and experiments.

### 5.2.2 Unsupervised Co-part Segmentation

Unsupervised co-part segmentation aims to discover and segment semantic parts for an object. It indicates whether the content representations are semantically meaningful. Existing methods (Hung et al., 2019; Liu et al., 2021) rely heavily on hand-crafted dataset-specific priors, such as background assumption and transformations. We choose SCOPS (Hung et al., 2019) and Liu et al. (2021) as baselines and follow the same setting to use landmark regression on Celeba-Wild dataset as a proxy task for evaluation. Please refer to the Appendix E.2 for details.

Table 4: Landmark regression results on CelebA-Wild. $K = V - 1$ indicates the number of foreground parts.

| Method | $K = 4$ | $K = 8$ |
|---|---|---|
| SCOPS (w/o saliency) | 46.62 | 22.11 |
| SCOPS (with saliency) | 21.76 | 15.01 |
| Liu et al. (2021) | 15.39 | 12.26 |
| Retriever | **13.54** | **12.14** |

The content code of Retriever is visualized in Figure 6 (a) for the Celeba-Wild dataset and in Figure 8 for the DeepFashion dataset. By setting a small $V$ in Retriever, the encoded content only preserves the basic shape information as segmentation maps. Each content token is consistently aligned with a meaningful part among different images. When compared to the state-of-the-art

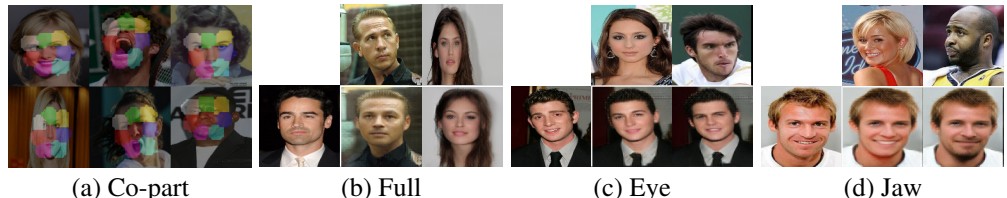

| (a) Co-part | (b) Full | (c) Eye | (d) Jaw |

Figure 6: Co-part segmentation and style transfer results on Celeba-Wild. Our method achieves desired appearance transfer even on the unaligned dataset.

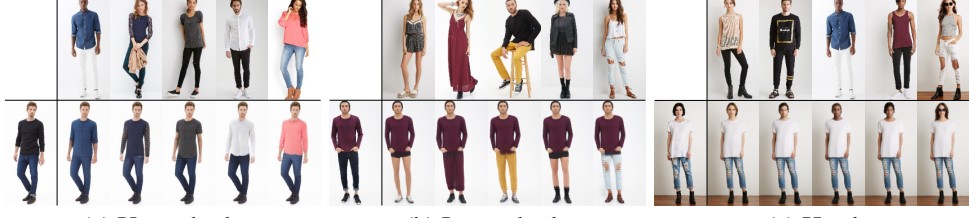

| (a) Upper body | (b) Lower body | (c) Head |

Figure 7: Part-level appearance manipulation on DeepFashion. Our method can retrieve the correct appearance even with occlusion and large deformation.

methods, as Table 4 shows, `Retriever` is one of the top performers on this task, even without any hand-crafted transformations or additional saliency maps.

### 5.2.3 UNSUPERVISED PART-LEVEL STYLE TRANSFER

Part-level style transfer requires disentanglement of shape and appearance at part-level, which is particularly challenging in an unsupervised manner. Previous works (Lorenz et al., 2019; Liu et al., 2021) propose to use one-one matching between shape and appearance, which is not flexible and leads to unrealistic results. Our content-style bipartite graph separates content and style at part-level and enables a more flexible content-specific style modeling. We can perform part-level style transfer by calculating the co-occurrence map between content and style tokens. See Appendix B for more details.

Both image-level and part-level transfer results are shown in Figure 6, 7, and 8. `Retriever` is capable of the explicit control of local appearance. Even without any supervision for the highly unaligned image pairs, our method can transfer the appearance to a target shape with high visual quality. See Appendix E.4 for more results.

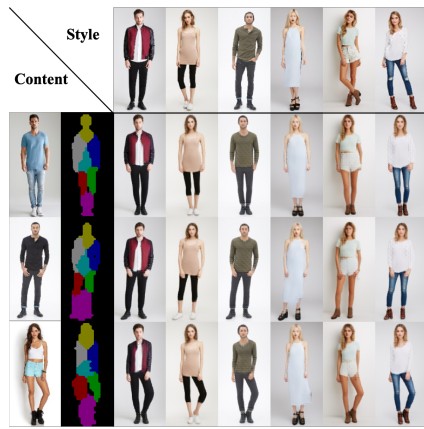

Figure 8: Shape and appearance transferring on DeepFashion.

## 6 CONCLUSION

In this paper, we have designed and implemented an unsupervised framework for learning separable and interpretable content-style features. At the core of the proposed `Retriever` framework are the two retrieval operations powered by the innovative use of multi-head attention. Cross-attention is used as the style encoder to retrieve style from the input structured data, and link attention is used as the decoder to retrieve content-specific style for data reconstruction. We have demonstrated that structural constraints can be integrated into our framework to improve the interpretability of the content. This interpretability is further propagated to the style through the innovative bipartite graph representation. As a result, the proposed `Retriever` enables a couple of fine-grained downstream tasks and achieves superior performance. As for the limitations, we have discovered in experiments that different tasks on different datasets need different model settings. We currently lack theoretical guidance for determining these settings, and we plan to work on it in the future.

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

## A  BACKGROUND

**Attention.** Given $n$ query vectors, each with dimension $d_q$: $\boldsymbol{Q} \in \mathbb{R}^{n \times d_q}$, an attention function $\text{Attn}(\boldsymbol{Q}, \boldsymbol{K}, \boldsymbol{V})$ maps queries $\boldsymbol{Q}$ to outputs using $m$ key-value pairs $\boldsymbol{K} \in \mathbb{R}^{m \times d_q}, \boldsymbol{V} \in \mathbb{R}^{m \times d_v}$,

$$\text{Attn}(\boldsymbol{Q}, \boldsymbol{K}, \boldsymbol{V}) = \text{softmax}(\frac{\boldsymbol{Q}\boldsymbol{K}^T}{\sqrt{d_q}})\boldsymbol{V},$$

where dot product $\boldsymbol{Q}\boldsymbol{K}^T \in \mathbb{R}^{n \times m}$ measures the similarity between queries and keys. $\text{softmax}(\cdot)$ normalizes this similarity matrix to serve as the weight of combining the $m$ elements in $\boldsymbol{V}$.

**Multi-Head Attention.** Vaswani et al. (2017) propose multi-head attention, denoted as $\text{MHA}(\cdot, \cdot, \cdot)$. It projects $\boldsymbol{Q}, \boldsymbol{K}, \boldsymbol{V}$ into $h$ different groups of query, key, and value vectors, and applies attention function on each group. The output is a linear transformation of the concatenation of all attention outputs:

$$\boldsymbol{O}_i = \text{Attn}(\boldsymbol{Q}\boldsymbol{W}_i^Q, \boldsymbol{K}\boldsymbol{W}_i^K, \boldsymbol{V}\boldsymbol{W}_i^V),$$

$$\text{MHA}(\boldsymbol{Q}, \boldsymbol{K}, \boldsymbol{V}) = \text{concat}(\boldsymbol{O}_1, \boldsymbol{O}_2, ..., \boldsymbol{O}_h)\boldsymbol{W}^O,$$

where $\boldsymbol{W}_i^Q, \boldsymbol{W}_i^K \in \mathbb{R}^{d_q \times \frac{d_q}{h}}$, and $\boldsymbol{W}_i^V \in \mathbb{R}^{d_v \times \frac{d_v}{h}}$ are the projection matrix for query, key, and value in the $i$-th head, $\boldsymbol{W}^O \in \mathbb{R}^{d_v \times d_q}$ is the linear transformation that fuses all the heads' outputs.

**Self-Attention.** Given an input array $\boldsymbol{X} \in \mathbb{R}^{n \times d_x}$, a self-attention $\text{SA}(\cdot)$ or multi-head self-attention $\text{MHSA}(\cdot)$ operation takes $\boldsymbol{X}$ as query, key, and value:

$$\text{SA}(\boldsymbol{X}) = \text{Attn}(\boldsymbol{X}, \boldsymbol{X}, \boldsymbol{X})$$

$$\text{MHSA}(\boldsymbol{X}) = \text{MHA}(\boldsymbol{X}, \boldsymbol{X}, \boldsymbol{X})$$

**Cross-Attention.** Given input arrays $\boldsymbol{X} \in \mathbb{R}^{n \times d_x}, \boldsymbol{Y} \in \mathbb{R}^{m \times d_y}$, a cross-attention $\text{CA}(\cdot, \cdot)$ or multi-head cross-attention $\text{MHCA}(\cdot, \cdot)$ operation takes $\boldsymbol{X}$ as query, and takes $\boldsymbol{Y}$ as key and value:

$$\text{CA}(\boldsymbol{X}, \boldsymbol{Y}) = \text{Attn}(\boldsymbol{X}, \boldsymbol{Y}, \boldsymbol{Y})$$

$$\text{MHCA}(\boldsymbol{X}, \boldsymbol{Y}) = \text{MHA}(\boldsymbol{X}, \boldsymbol{Y}, \boldsymbol{Y})$$

Property: (multi-head) cross-attention operation is P.I. to $\boldsymbol{Y}$, that is,

$$\forall \pi \in \mathcal{P}^m, \text{MHCA}(\boldsymbol{X}, \pi(\boldsymbol{Y})) = \text{MHCA}(\boldsymbol{X}, \boldsymbol{Y}),$$

where $\mathcal{P}^m$ is the set of all permutations of indices $\{1, ..., m\}$

**Permutation Invariant Property of Style Encoder** For the $i$-th style encoder block, we assume its input $\boldsymbol{Z}_{i-1}$ is P.I. to $\boldsymbol{F}$. The output of the multi-head cross-attention layer is $\hat{\boldsymbol{Z}}_i = \text{MHCA}(\boldsymbol{Z}_{i-1}, \boldsymbol{F})$. According to the property of multi-head cross-attention, we have $\hat{\boldsymbol{Z}}_i$ is permutation invariant to $\boldsymbol{F}$. The following residue connection combines $\hat{\boldsymbol{Z}}_i$ with $\boldsymbol{Z}_{i-1}$, both of which are P.I. to $\boldsymbol{F}$. Thus the combination is P.I. to $\boldsymbol{F}$. The following token mixing layer and feed-forward layer don't take $\boldsymbol{F}$ as input. Because their inputs are P.I. to $\boldsymbol{F}$, their outputs are also P.I. to $\boldsymbol{F}$. To summarize, if $\boldsymbol{Z}_{i-1}$ is P.I. to $\boldsymbol{F}$, then $\boldsymbol{Z}_i$ is also P.I. to $\boldsymbol{F}$. It is obvious that $\boldsymbol{Z}_0$ is P.I. to $\boldsymbol{F}$ since $\boldsymbol{Z}_0$ is fixed during the forward path. Therefore, the output of the whole style encoder is P.I. to $\boldsymbol{F}$.

**Vector Quantization.** For efficiency, we use product quantization (Baevski et al., 2020a;b). The quantized feature $\boldsymbol{c}_i$ is the concatenation of $G$ embeddings $\boldsymbol{e}_1, \boldsymbol{e}_2, ..., \boldsymbol{e}_G \in \mathbb{R}^{d_c/G}$, which are looked up from the corresponding $G$ codebooks $\boldsymbol{E}_1, \boldsymbol{E}_2, ..., \boldsymbol{E}_G \in \mathbb{R}^{V \times d_c/G}$, where $V$ is the number of entries in each codebook. To make VQ operation differentiable, Gumbel-softmax (Jang et al., 2017) is adopted: Firstly, each input token is mapped to logits $\boldsymbol{l} \in \mathbb{R}^{G \times V}$. Then, $\boldsymbol{e}_g$ is calculated by a weighted sum of codewords in codebook $\boldsymbol{E}_g$, and the weight for adding the $v$-th code in group $g$ is given by:

$$w_{g,v} = \frac{\exp((l_{g,v} + n_{g,v})/\tau)}{\sum_{k=1}^{V} \exp((l_{g,k} + n_{g,k})/\tau)}, \tag{2}$$

where $\tau$ is temperature, $n_{g,v} = -\log(-\log(u_{g,v}))$, and $u_{g,v} \sim \mathcal{U}(0, 1)$ is independently sampled for all subscripts. We use the straight-through estimator (Jang et al., 2017), treating the forward and backward path differently. During the forward path, the one-hot version of the above weight

$one\_hot(\arg\max_k w_{g,k})$ is used, and in the backward path, the gradient of the original $w_{g,k}$ is used. To encourage all the VQ codes to be equally used, we use a batch-level VQ perplexity loss:

$$\mathcal{L}_{VQ} = \frac{1}{GV}\sum_{g=1}^{G}\exp(-H(\overline{\boldsymbol{p}}_g)) = \frac{1}{GV}\sum_{g=1}^{G}\exp(\sum_{v=1}^{V}\overline{p}_{g,v}\log\overline{p}_{g,v})), \qquad (3)$$

where $\overline{\boldsymbol{p}}_g$ indicates the probability calculated by $\text{softmax}(\boldsymbol{l}_g)$ averaged across all the tokens in batch, $H(\cdot)$ calculates the entropy of a discrete distribution.

## B  VISUALIZATION OF LEARNED CONTENT-STYLE BIPARTITE GRAPH

`Retriever` models the content-style bipartite graph by introducing the novel link attention module. The graph's edges are represented by the link attention map. To understand what content is linked to which style, we gather the statistics of the link attention map as the content-style co-occurrence map. To show content information in a 2-d map, the content is categorized by either the ground-truth labels, or the discovered parts or units. Specifically, the co-occurrence statistics of content category $c$ and style token $s$, denoted as $m_{c,s}$, is calculated as follows:

$$m_{c,s} = \frac{1}{|\mathcal{D}|}\sum_{x\in\mathcal{D}}A(x,s)I(l(x)=c),$$

where $\mathcal{D}$ indicates the set composed of all the content tokens in the target dataset. $A(x,s)$ is the link attention amplitude between content token $x$ and style token $s$. $I(\cdot)$ is the characteristic function. It outputs 1 if its input is true and outputs 0 if its input is false. $l(x)$ gives the content category of token $x$.

For each style token, we consider the content category that shows the strongest co-occurrence with it to be the major content category that it serves. For the convenience of finding the pattern inside the co-occurrence map, we normalize on both content category axis and style token axis, then sort the style tokens so that their major content category index is in the ascending order.

In audio domain, we categorize content tokens using ground-truth phoneme labels and calculate the co-occurrence map using the first link attention. The content-style co-occurrence map is shown in Figure 9.

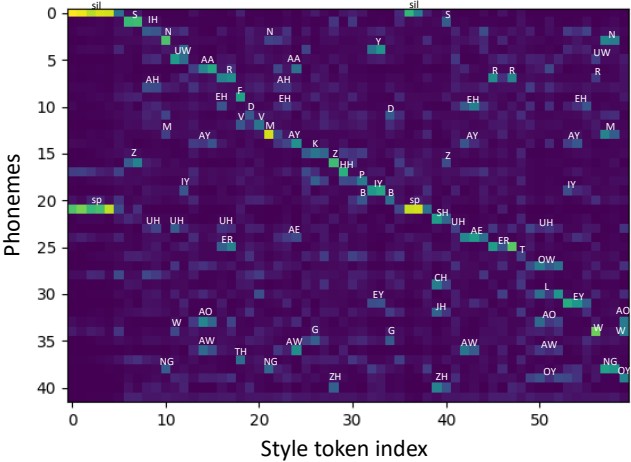

Figure 9: Phoneme-style co-occurrence map of higher resolution.

In image domain, we categorize content tokens using the discovered parts and calculate the co-occurrence map using the third link attention. The part indices together with content-style co-occurrence map are shown in Figure 10.

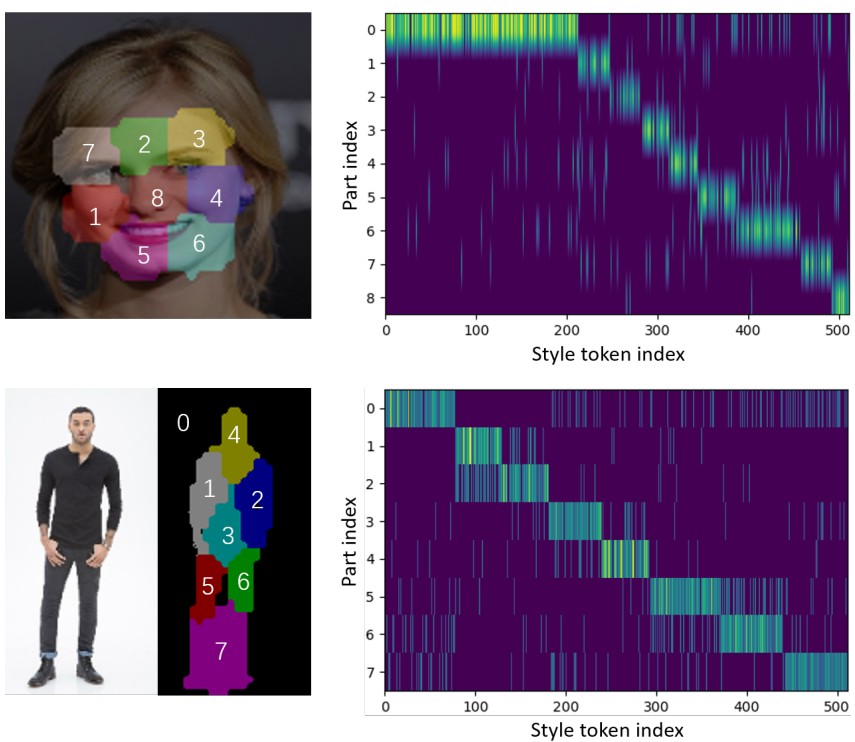

Figure 10: Visualization of part-style co-occurrence map on Celeba-Wild and DeepFashion dataset.

## C    STYLE SPACE VISUALIZATION

In order to provide an intuitive impression about what the styles decomposed by our framework are, we present visualizations of some style spaces produced by t-SNE (Van der Maaten & Hinton, 2008). We see the clustered patterns in many style spaces which correspond to meaningful attributes like person gender and clothes color.

### C.1    SPEECH DOMAIN

We visualize the style vectors extracted from 500 randomly-selected utterances of ten unseen speakers from Librispeech test-clean split. In Figure 11, we show the latent space of four typical style vectors, which correspond to the style of nasal consonant (NG), voiceless consonant (T), voiced consonant (Z), and vowel (IH). Different speakers are labeled in different colors. We see that different speakers are clustered into different clusters.

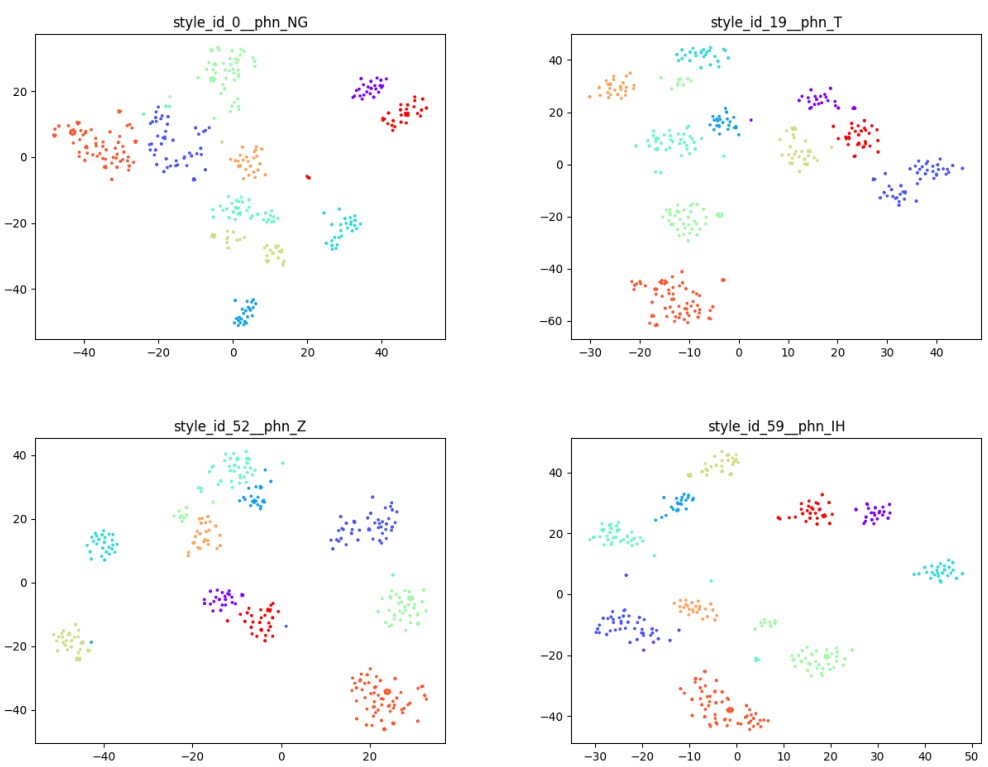

Figure 11: Style space t-SNE visualization in speech domain: utterances of the same speaker cluster together.

### C.2    IMAGE DOMAIN

We examine the style vector spaces on Deepfashion dataset, and find that some style vector spaces are separately or collectively correlated with the human-defined attributes. For example, we find that the style vector #43 is highly correlated with the color of the clothes. Figure 12 shows the t-SNE visualization of this vector space, as well as some images in each cluster. It is clear that images are clustered solely based on the color of the clothes despite of various model identifications, standing poses, and styles of the dress.

As this dataset also has the gender attribute labelled, we further look for the 'gender vector' in the decomposed style vector spaces. Interestingly, we find 14 vector spaces that are highly correlated to the gender attribute. We concatenate these 14 vectors and show the t-SNE visualization in Figure 13.

We can see a clear boundary between male and female models despite of the their color of clothes or standing poses.

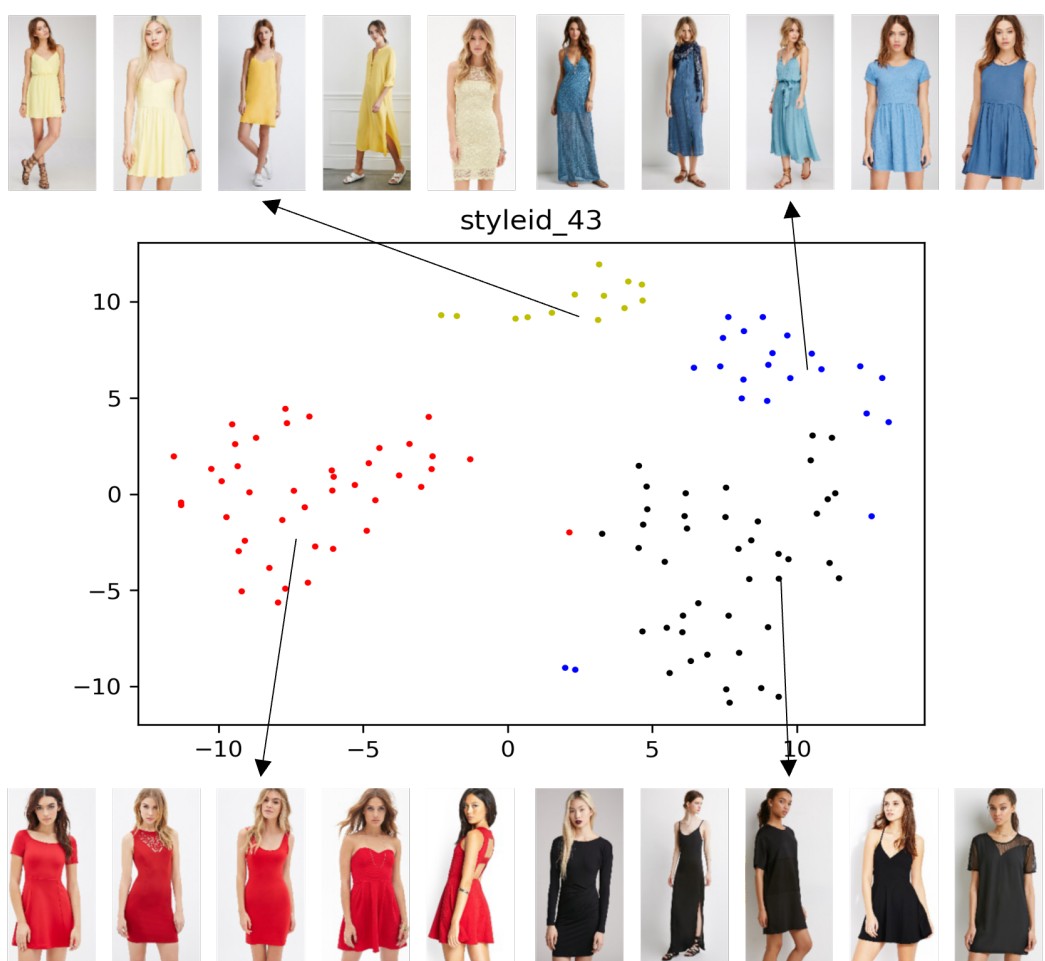

Figure 12: Style space t-SNE visualization in image domain: style vector #43 is a 'clothes color vector'.

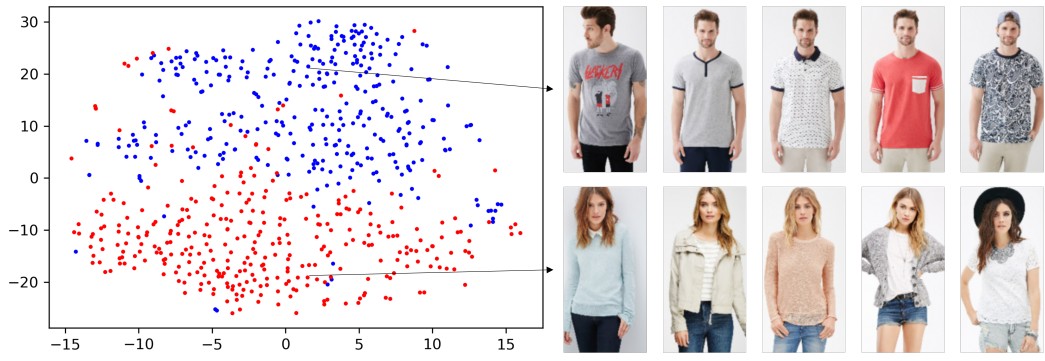

Figure 13: Style space t-SNE visualization in image domain: an ensemble of 14 style vectors identifies gender.

## D    Structural Constraint

### D.1    Image domain

To make the content representation interpretable, we can apply additional structural constraint. Inspired by the observation that pixels belonging to the same object part are usually spatially concentrated and form a connected region, we follow Hung et al. (2019) to use a geometrical concentration loss as the structural constraint to shape the content representation to part segments. In practice, we denote the pixels assigned to the first entry in the VQ codebook as background pixels and the other pixels as the foreground pixels.

Given the logits $L \in \mathbb{R}^{V \times H \times W}$ that assign each pixel to the parts (entries in VQ codebook), for foreground pixels:

$$\mathcal{L}_{fg} = \frac{1}{V-1} \sum_{v=2}^{V} \sum_{h,w} \left\| \begin{bmatrix} h \\ w \end{bmatrix} - \begin{bmatrix} c_h^v \\ c_w^v \end{bmatrix} \right\|_2^2 \cdot L_{v,h,w}/z_v, \tag{4}$$

where $c_h^k$ and $c_w^k$ are the coordinates of the $v$-th part center along axis $h$, $w$ and $z_k$ is a normalization term. $c_h^k$, $c_w^k$ and $z_k$ can be calculated as follows:

$$c_h^v = \sum_{h,w} h \cdot L_{v,h,w}/z_v, c_w^v = \sum_{h,w} w \cdot L_{v,h,w}/z_v, \tag{5}$$

$$z_v = \sum_{h,w} L_{v,h,w}. \tag{6}$$

For the background pixels, different from previous methods using background loss (Liu et al., 2021) or saliency map to bound the region of interested object, we do not add any constraint. Thus $\mathcal{L}_{SC} = \mathcal{L}_{fg}$

### D.2    Audio domain

To discover phoneme-like units in speech, the discrete representation should not switch too frequently along the time axis. Therefore, we penalize the code switch based on the calculation of neighborhood cross-entropy.

$$\mathcal{L}_{CE} = \frac{1}{N-1} \sum_{i=1}^{N-1} \text{CE}(\boldsymbol{p}_i, z_{i+1}) + \text{CE}(\boldsymbol{p}_{i+1}, z_i), \tag{7}$$

where $N$ is the length of the input sequence, $\text{CE}(\cdot)$ calculates the cross-entropy between discrete probability and given label, $\boldsymbol{p}_i$ denotes the predicted discrete probability before Gumbel-softmax, and $z_i$ is the highest-probability code on position $i$.

However, $\mathcal{L}_{CE}$ also penalizes the true phoneme switch, where the input acoustic feature really changes dramatically. To solve this problem, we use a truncated version of $\mathcal{L}_{CE}$ as our structural constraint:

$$\mathcal{L}_{SC} = \gamma \tanh(\frac{\mathcal{L}_{CE}}{\gamma}), \tag{8}$$

where $\gamma$ is a hyper-parameter indicating the truncation threshold. For the true phoneme switch, high $\mathcal{L}_{CE}$ is expected, thus the loss will be truncated, while within the same phoneme, low $\mathcal{L}_{CE}$ is expected, and $\mathcal{L}_{SC}$ behaves like the original CE loss. As such, the truncated loss is more suitable for discovering phoneme-like units. In our experiment, we set $\gamma = \ln(V) = \ln(100)$.

## E    Image: Additional Details

### E.1    Implementation Details

We first show the tokenization module in Table 5 and the detokenization module in Table 6. For the layer numbers, we set $L_e, L_s, L_d$ to $6, 3, 6$, respectively. The hyper-parameter settings of each style-encoder layer and decoder layer are shown in Table 7 and Table 8, where $d_{FFN}$ indicates the hidden

dimension of the feed-forward network, and $n_{head}$ is the number of heads used in the multi-head attention modules. Specifically, for the decoder, we replace the FFN with Mix-FFN to improve the image quality. The Mix-FFN can be formulated as:

$$x_{out} = \text{MLP}(\text{GELU}(\text{Conv}_{3\times3}(\text{MLP}(x_{in})))),$$

where $x_{in}$ is the input feature. The Gumbel-Softmax temperature $\tau$ in VQ is annealed from 2 to a minimum of 0.01 by a factor of 0.9996 at every update.

For the Celeba-Wild dataset, we resize the input to $128 \times 128$. For DeepFashion dataset, we resize the input to $192 \times 96$. For the number of style tokens, we use 512. We summarize the training setting in Table 9. Our model is implemented with Pytorch and trained on 4 Nvidia V100 GPUs.

Table 5: Tokenization module for images.

| |
|---|
| Conv $3 \times 3 \times 3 \times 24$, `stride = 1` |
| BatchNorm |
| ReLu |
| Conv $3 \times 3 \times 24 \times 48$, `stride = 2` |
| BatchNorm |
| ReLu |
| Conv $3 \times 3 \times 48 \times 96$, `stride = 1` |
| BatchNorm |
| ReLu |
| Conv $3 \times 3 \times 96 \times 192$, `stride = 2` |
| BatchNorm |
| ReLu |
| Conv $3 \times 3 \times 192 \times 192$, `stride = 1` |

Table 6: Detokenization module for images.

| |
|---|
| PixelShuffle(2) |
| Conv $3 \times 3 \times 48 \times 48$, `stride = 1` |
| ReLu |
| PixelShuffle(2) |
| Conv $3 \times 3 \times 12 \times 12$, `stride = 1` |
| ReLu |
| Conv $1 \times 1 \times 12 \times 3$, `stride = 1` |

Table 7: Style encoder hyper-parameters.

| Hyper-parameter | Value |
|---|---|
| $d_s$ | 192 |
| $d_{FFN}$ | 768 |
| $n_{head}$ | 4 |

Table 8: Decoder hyper-parameters.

| Hyper-parameter | Value |
|---|---|
| $d$ & $d_c$ | 192 |
| $d_{FFN}$ | 768 |
| $n_{head}$ | 4 |

Table 9: Training setting.

| Hyper-parameter | Value |
|---|---|
| $\lambda_{rec}$ | 1 |
| $\lambda_{VQ}$ | 0.3 |
| $\lambda_{sc}$ | 4 or 7 |
| optimizer | Adam ($\beta_1 = 0.9, \beta_2 = 0.999$) |
| Learning rate | 0.001 |
| Batchsize | 16 |
| Iteration | 150,000 |

### E.2 DETAILS ABOUT EVALUATION ON CO-PART SEGMENTATION

For the evaluation on co-part segmentation, we follow the setting in Liu et al. (2021) and Hung et al. (2019). The proxy task for Celeba-Wild is landmark regression. We first convert part segmentations into landmarks by taking part centers as in Eq. 5. Then we learn a linear regressor to map the converted landmarks to ground-truth landmarks and evaluate the regression error on test data.

E.3 ABLATION STUDY

Table 10: Ablation study on CelebA-Wild.

| Method | Landmark regression error |
|---|---|
| Ours w/ bipartite graph | **12.14** |
| Ours w/ AdaIN | 13.07 |
| Ours w/o style branch | 15.62 |
| Ours w/ $16\times$ downsampling rate | 44.89 |
| Ours w/ $8\times$ downsampling rate | 30.62 |
| Ours w/ $4\times$ downsampling rate | **12.14** |
| Ours w/ $\lambda_{sc} = 4$ | 19.32 |
| Ours w/ $\lambda_{sc} = 7$ | **12.14** |
| Ours w/ $\lambda_{sc} = 10$ | 16.75 |
| Ours w/ $\lambda_{vq} = 0$ | 15.98 |
| Ours w/ $\lambda_{vq} = 0.01$ | 13.58 |
| Ours w/ $\lambda_{vq} = 0.1$ | 12.84 |
| Ours w/ $\lambda_{vq} = 0.3$ | **12.14** |
| Ours w/ $\lambda_{vq} = 1$ | 21.60 |
| Ours w/ Conv | **12.14** |
| Ours w/ PatchMerge | 17.16 |

**Do the style encoder and decoder impact the co-part segmentation?** In this work, we find VQ module is suitable for unsupervised co-part segmentation, as it is a cluster center among the whole dataset. Here we evaluate that if the style branch further helps the co-part segmentation. As shown in Table 10, with a style branch, the discovered part segmentation is more consistent. Moreover, we compare our link attention decoder with AdaIN operation and find that our design is better for co-part segmentation task. This further verified that our content-style bipartite graph is a powerful representation with high interpretability.

**On using different structural constraint weight.** We further carry out experiments with different loss weight for structural constraint. Without the structural constraint, the parts learned are scattered all over an image. As the weight for structural constraint getting larger, the part segmentation becomes tighter. With too large weight, the regions lose flexibility and fail to cover the parts.

**On using different VQ diversity loss weight.** We also test our model's sensitivity to $\lambda_{vq}$. As long as $\lambda_{vq}$ is not set to zero or too large (e.g., $\lambda_{vq} = 1$), the model has a fairly stable landmark regression error even when $\lambda_{vq}$ is adjusted from 0.01 to 0.3.

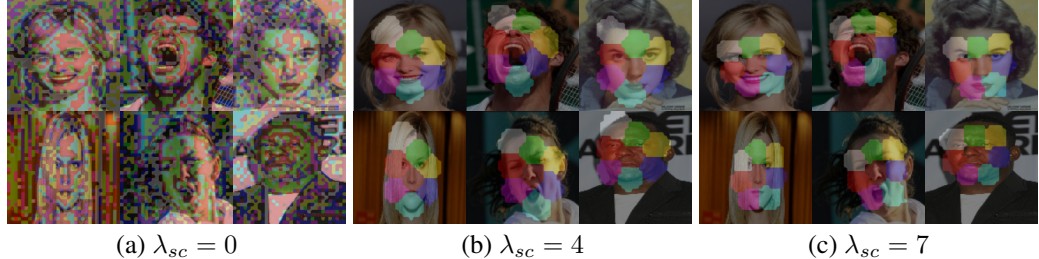

(a) $\lambda_{sc} = 0$         (b) $\lambda_{sc} = 4$         (c) $\lambda_{sc} = 7$

Figure 14: Visual ablation study on weight of structural constraint $\lambda_{sc}$.

**On using different tokenization module.** Different tokenization modules bring additional bias. Besides a stack of convolution layers to tokenize the input image, we also experiment with the PatchMerge operation in ViT (Dosovitskiy et al., 2021), which is a $4 \times 4$ convolution layer with stride 4 in our case. As shown in Table 10, using convolution layers as the tokenization module can help the preprocessing transformer blocks inside `Retriever` work better, which is also observed by a concurrent work (Xiao et al., 2021).

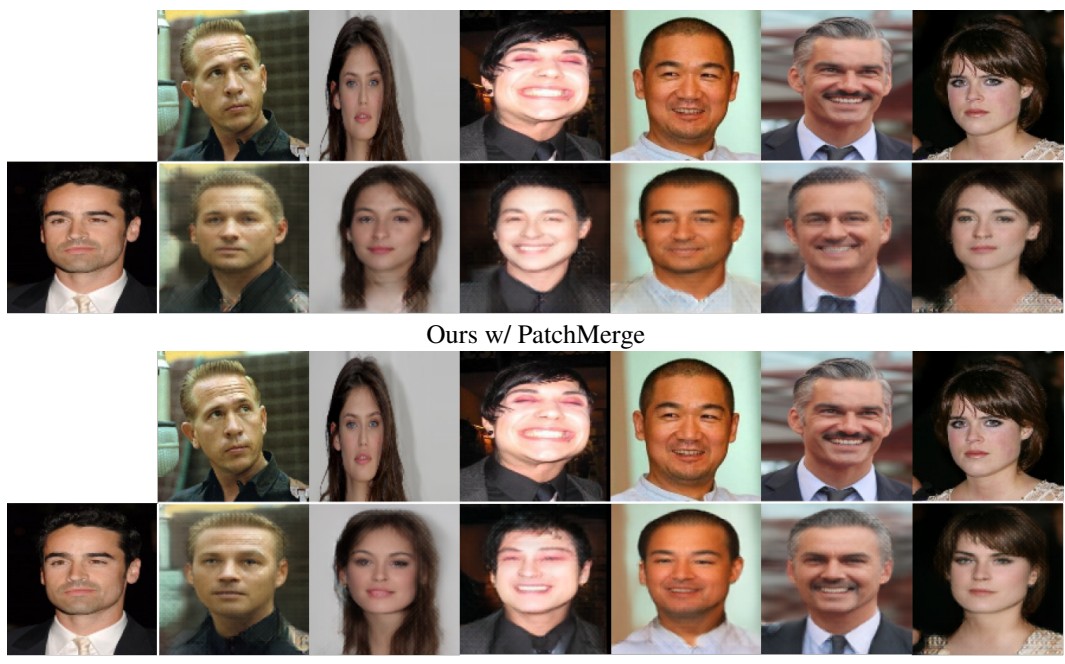

Figure 15: Visual ablation study on tokenization module. Tokenizing the images using convolutions can help our model extract the content and style better, which leads to better transfer results.

**On using different downsampling rate.** In this work, we mainly use $4\times$ downsampling rate. For the input $128 \times 128$ image, we first tokenize it to $32 \times 32$. Here we experiment with additional different downsampling rates: $8\times$ and $16\times$. In addition to the convolution layer for $4\times$ setting, we add additional convolution layers for further downsampling. We do not try $2\times$ due to the limited computational resource. As shown in Table 10, for the selected downsampling rate, larger one results in worse performance. We also provide visual comparison in Figure 16.

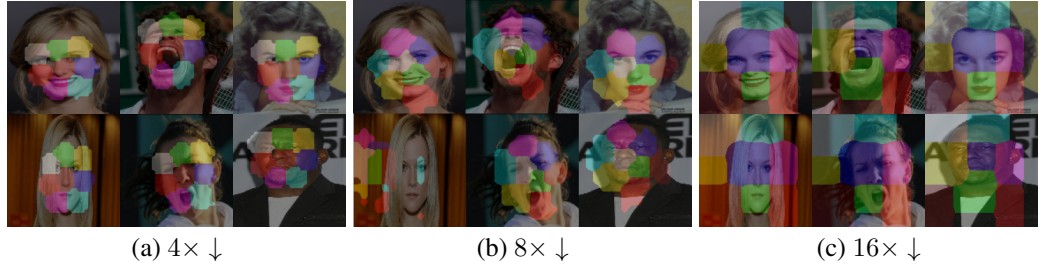

(a) $4\times \downarrow$ (b) $8\times \downarrow$ (c) $16\times \downarrow$

Figure 16: Visual ablation study on downsampling rate. Larger downsampling rate leads to coarser co-part segmentation results.

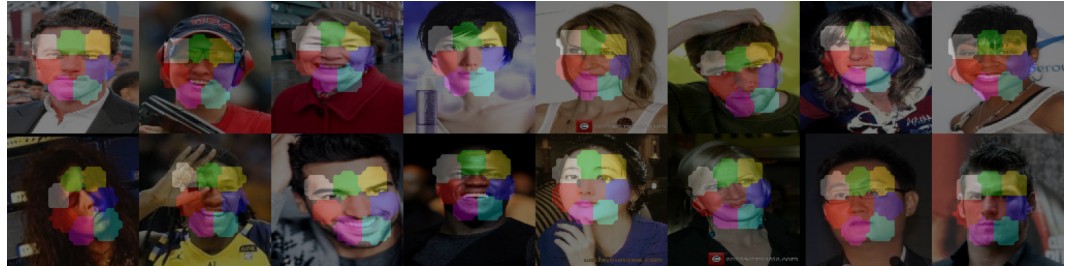

Figure 17: More co-part segmentation results on Celeba-Wild dataset.

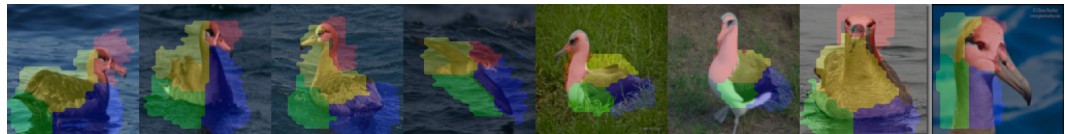

Figure 18: Co-part segmentation results on CUB dataset. We set $K = 4$. Similar to previous work, due to the geometric concentration loss, we find it struggles with distinguishing between orientations of near-symmetric objects, which is a common limitation.

### E.4  MORE QUALITATIVE RESULTS

**Co-part segmentation.** We first provide more visualization on Celeba-Wild in Figure 17. Besides, we also apply our method on CUB dataset (Wah et al., 2011). CUB is a dataset with 11, 788 photos of 200 bird species in total. This dataset is challenging due to the pose of the birds is of large diversity. We follow (Liu et al., 2021) select the photos from the first 3 categories, and get about 200 photos. The results are shown in Figure 18.

**Part-level style transfer.** Besides the results in the main paper, we also apply our method on a high-resolution dataset: CelebA-HQ. We resize the images to $256 \times 256$. We show the part-level style transfer results in Figure 19 for mouth, nose, and eye.

**Zero-shot image style transfer.** We find our model capable of generalizing into unseen image datasets. In Figure 20, we use our model trained on CelebA-Wild to transfer artistic style from an artistic image. The transfer result keeps the pose and shape of the content images (the first row), while adopting the tone and appearance of the style image (the first column on the left).

**Visual comparison.** We find our method enables more natural style transfer compared with the previous work. Here we show side-by-side visual comparison on Deepfashion dataset in Figure 21 and 22. In image-level style transfer, the results of Lorenz et al. (2019) suffer from significant deformation artifact. In contrast, such artifact is not observed in our results. In part-level style transfer, we show the results of head transfer. Again, our results are more natural and contain more image details.

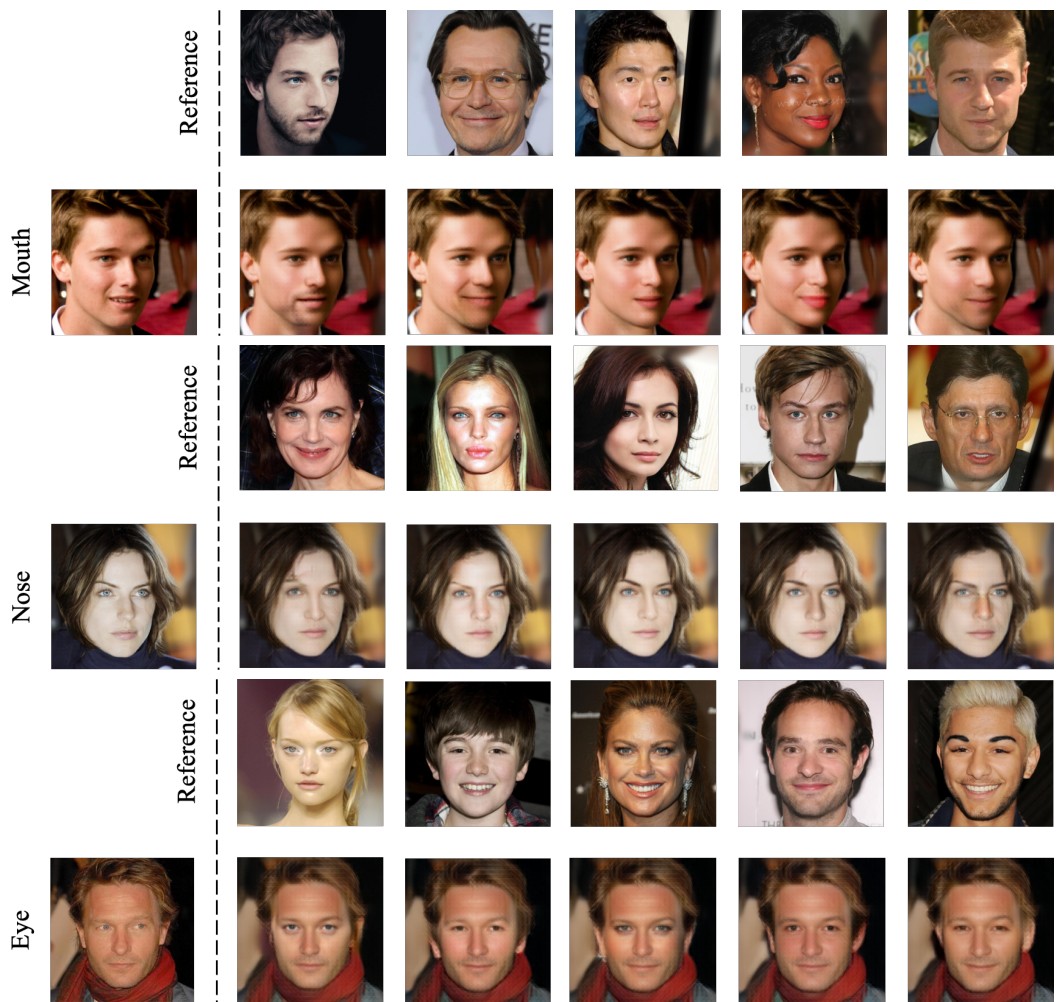

Figure 19: Part-level style transfer results on CelebA-HQ dataset for mouth, nose and eye. The resolution is $256 \times 256$.

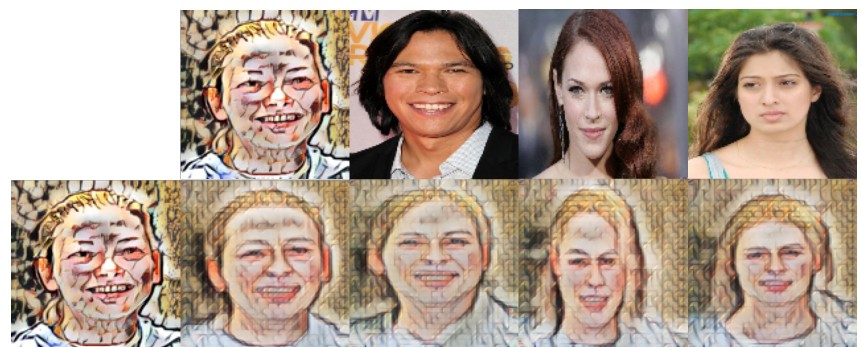

Figure 20: Zero-shot image style transfer. Our model is trained on CelebA-Wild but can be generalized to artistic images. The artist image (left) is obtained by AdaIN (Huang & Belongie, 2017) from CelebA-Wild (Liu et al., 2015) and WikiArt (Saleh & Elgammal, 2015) dataset.

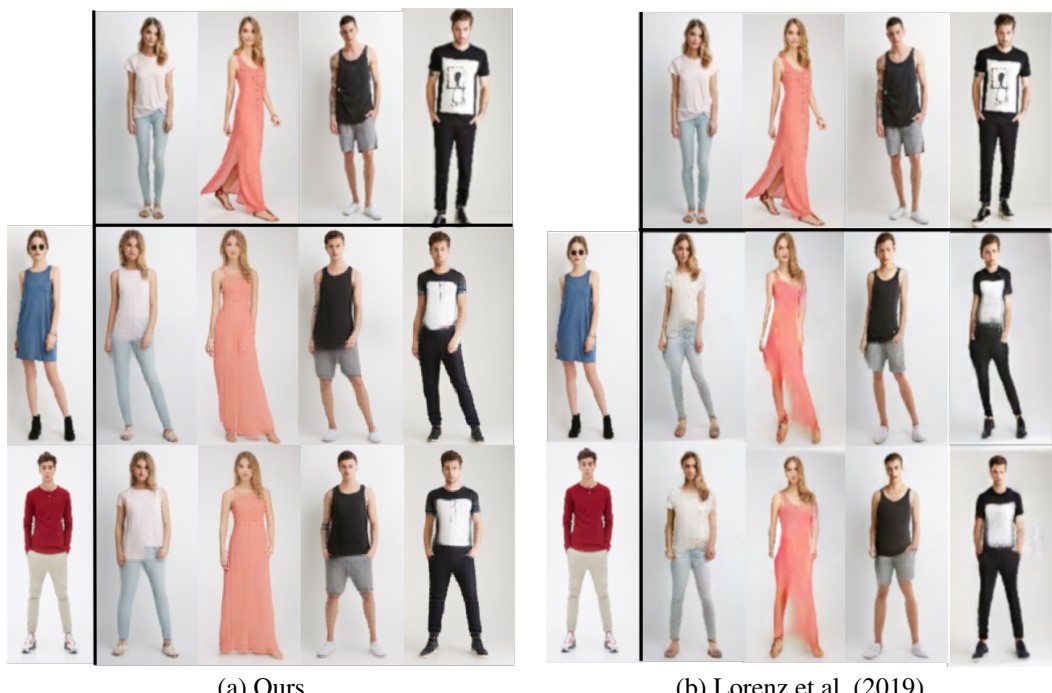

(a) Ours             (b) Lorenz et al. (2019)

Figure 21: Visual comparison on Deepfashion dataset: image-level style transfer. Our results are more natural than the baseline. Results of Lorenz et al. (2019) are cropped from their official website: `https://compvis.github.io/unsupervised-disentangling`.

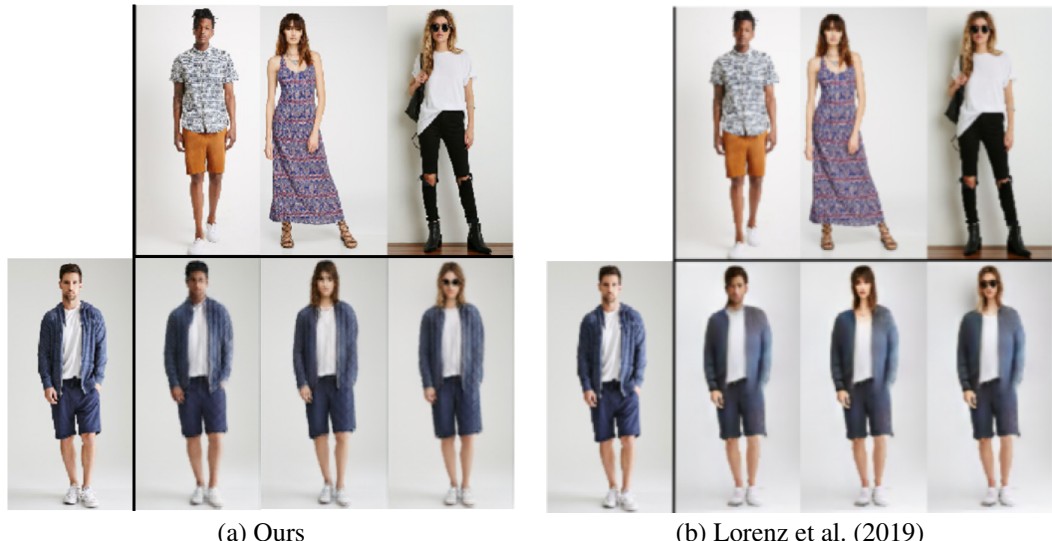

(a) Ours             (b) Lorenz et al. (2019)

Figure 22: Visual comparison on Deepfashion dataset: head style transfer. Our results are more natural than the baseline and contain more image details.

## F   AUDIO: ADDITIONAL DETAILS

### F.1   IMPLEMENTATION DETAILS

For the tokenization module, we follow Lin et al. (2021a), using s3prl toolkit[1] to extract the CPC feature, and use a depth-wise convolutional layer with kernel size 15 after the CPC feature. The depth-wise convolutional layer is trainable, while the CPC model is fixed during training. For the detokenization module, we use publicly available implementation of Parallel WaveGAN[2] pretrained on LibriTTS. $L_e, L_s, L_d$ are set to 0, 3, 4, respectively. The dimension of content representation is $d_c = 512$. Following Qian et al. (2020), we add auxiliary quantized normalized $f_0$ feature into content for better voice quality. The Gumbel-Softmax temperature $\tau$ is annealed from 2 to a minimum of 0.01 by a factor of 0.9996 at every update. The hyper-parameter settings of each style-encoder layer and decoder layer are shown in Table 11 and Table 12, where $d_{FFN}$ indicates the hidden dimension of the feed-forward network, and $n_{head}$ is the number of heads used in the multi-head attention modules. The decoder output is converted to 80-d log-Mel spectrum with an MLP whose hidden dimension is 4096. The training hyper-parameter setting is listed in Table 13. Each training sample is a random 4-second audio clip sampled from the training dataset. For utterances that are shorter than 4s, we pad zero at the end of the utterance. The code is implemented with Pytorch, and training takes 5 hours on 4 Nvidia V100 GPUs.

Table 11: Style encoder hyper-parameters.

| Hyper-parameter | Value |
| --- | --- |
| $d_s$ | 192 |
| $d_{FFN}$ | 512 |
| $n_{head}$ | 4 |
| dropout | 0.1 |

Table 12: Decoder hyper-parameters.

| Hyper-parameter | Value |
| --- | --- |
| $d$ | 512 |
| $d_{FFN}$ | 2048 |
| $n_{head}$ | 8 |
| dropout | 0.1 |

Table 13: Training setting.

| Hyper-parameter | Value |
| --- | --- |
| $\lambda_{rec}$ | 5 |
| $\lambda_{VQ}$ | 0.3 |
| $\lambda_{SC}$ | 0.1 |
| Optimizer | Adam ($\beta_1 = 0.9, \beta_2 = 0.999$) |
| Learning rate | 0.004 |
| Learning rate schedule | power ($p = 0.3$, warmup-steps = 625 ) |
| batch-size | 120 |
| epoch | 50 |

### F.2   EVALUATION METRICS OF ZERO-SHOT VC TASK

For objective metrics, conversion similarity is measured by Resemblyzer speaker verification system [3] by calculating speaker verification accuracy (SV Accu) between converted speech and corresponding target utterance, as done in the previous work (Lin et al., 2021b). The conversion is considered successful if the cosine similarity between Resemblyzer speaker embedding of converted speech and target utterance exceeds a pre-defined threshold. The threshold is decided based on equal error rate (EER) of the SV system over the whole testing dataset. The SV accuracy is the percentage of successful conversions. The objective speech quality metric is estimated by MOSNet (Lo et al., 2019). It takes converted speech as input and outputs a number ranging from 1 to 5 as the measurement of speech naturalness. Both metrics are higher the better.

---

[1] https://github.com/s3prl/s3prl
[2] https://github.com/kan-bayashi/ParallelWaveGAN
[3] https://github.com/resemble-ai/Resemblyzer

For subjective metrics, we conduct two tests to calculate the mean opinion score of conversion similarity (denoted as SMOS), and speech quality (denoted as MOS). For conversion similarity, each subject is asked to listen to the target utterance and converted utterance, and judge how confident they are spoken by the same speaker. For speech quality, each subject is asked to listen to utterances that are randomly chosen from converted speech and real speech, and determine how natural they sound. Both test results are scored from 1 to 5, the higher, the better. We randomly sample 40 utterances from the test set for both tests. Each utterance is evaluated by at least 5 subjects. The score is then averaged and reported with 95% confidence intervals.

### F.3    Phoneme information probing: training and testing

We conduct phoneme information probing to test how much phoneme information is encoded in the learned discrete content tokens, so as to demonstrate their interpretability. We experiment with two settings, one considers the contextual information (denoted as "Frame w/ context"), and the other considers only single-frame information (denoted as "Frame"). For these two settings, a 1d convolutional layer with kernel size 17 or a linear layer is used as the probing network respectively, to predict ground-truth phoneme label. For the input of the probing network, each frame is represented by the one-hot vector of the corresponding code. For the experiment involving two groups of code, the one-hot vector of both groups are concatenated on the channel axis. For the k-means clustering of CPC feature, the cluster number is set to 100, which is the same as the number of codebook entries per VQ group. We train the probing network on LibriSpeech train-clean-100 split and test it on LibriSpeech test-clean split. Adam optimizer is used with $\beta_1 = 0.9, \beta_2 = 0.999$. Learning rate and batch size are set to 0.00005 and 30, respectively. Each training and testing sample is a random 2-second segment from the dataset. We drop the utterance that is shorter than 2 seconds. In Table 1, we report the test accuracy after the convergence of the training.

### F.4    Details in system comparison

We choose four SOTA systems for comparison. Two content-style disentanglement approaches are AutoVC (Qian et al., 2019)[4], and AdaIN-VC (Chou & Lee, 2019)[5]. Two deep concatenative approaches are FragmentVC (Lin et al., 2021b)[6], and S2VC (Lin et al., 2021a)[7]. All these methods have officially released their code, together with the pretrained models also on VCTK dataset. We adopt these models and test them on our randomly sampled 1000 conversion pairs in CMU Arctic dataset.

### F.5    Inference scalability on zero-shot VC task

Table 14 shows the inference scalability of our method on VCTK + CMU Arctic dataset. "# target" indicates the number of available target utterances for inference. We see that `Retriever` performs quite well in the case that only one target utterance is given. As the available target utterance increases, the conver-

Table 14: Inference scalability

| # target | 1 | 2 | 3 | 5 | 10 |
|---|---|---|---|---|---|
| SV accu (%) | 95.1 | 98.7 | 99.4 | 99.4 | 99.6 |
| MOSNet | 3.12 | 3.13 | 3.12 | 3.12 | 3.13 |

sion similarity keeps increasing up to 99.6%, and it has no negative effect on speech quality, indicating the extracted style becomes more accurate when seeing more target samples.

### F.6    Performance on LibriSpeech dataset

LibriSpeech dataset (Panayotov et al., 2015) is more diverse than VCTK + CMU Arctic in the aspect of emotion, vocabulary, and speaker number, thus closer to real-world application. In this experiment, the whole train-clean-100 split containing 251 speakers is used for training, and test-clean split containing 40 speakers is used for testing. When testing, the conversion source-target pairs are built as follows: for each utterance in the test set, we treat it as the source utterance, and

---

[4]`https://github.com/auspicious3000/autovc`
[5]`https://github.com/jjery2243542/adaptive_voice_conversion`
[6]`https://github.com/yistLin/FragmentVC`
[7]`https://github.com/howard1337/S2VC`

assign it one target utterance that is randomly sampled from the test set. In this way, each utterance in the dataset serves as the source for one time.

Objective similarity metric is measured for using different style token numbers, as shown in Figure 23. The conversion similarity increases when using more style tokens, and the performance saturates at nearly 100%. To be specific, we achieve 98.4% SV accuracy and 3.13 MOSNet score when using 60 style tokens, demonstrating that our method can be generalized into more complicated scenarios and potentially can be used in the real-world application.

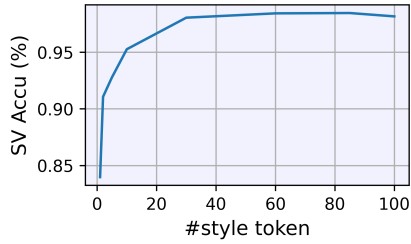

Figure 23: Performance on LibriSpeech dataset.

### F.7 LOSS WEIGHT ABLATION STUDY

Table 15: Loss weight ablation study on VCTK + CMU Arctic dataset.

| Method | SV Accu (%) | MOSNet |
|---|---|---|
| Ours w/ $\lambda_{SC} = 0$ | 98.9 | 3.01 |
| Ours w/ $\lambda_{SC} = 0.002$ | 98.5 | 3.10 |
| Ours w/ $\lambda_{SC} = 0.01$ | 99.0 | 3.11 |
| Ours w/ $\lambda_{SC} = 0.02$ | 98.5 | 3.12 |
| Ours w/ $\lambda_{SC} = 0.05$ | 99.0 | 3.10 |
| Ours w/ $\lambda_{SC} = 0.1$ | **99.4** | **3.12** |
| Ours w/ $\lambda_{SC} = 0.2$ | 99.7 | 3.07 |
| Ours w/ $\lambda_{SC} = 0.5$ | 99.3 | 3.12 |
| Ours w/ $\lambda_{VQ} = 0$ | 99.4 | 2.94 |
| Ours w/ $\lambda_{VQ} = 0.01$ | 99.5 | 3.08 |
| Ours w/ $\lambda_{VQ} = 0.3$ | **99.4** | **3.12** |
| Ours w/ $\lambda_{VQ} = 2.0$ | 99.5 | 3.08 |

To test the effectiveness of structural constraint and VQ diversity loss, we do ablation study on the corresponding loss weights, $\lambda_{VQ}$ and $\lambda_{SC}$. The SV accuracy and MOSNet scores achieved at different parameter settings are shown in Table 15. We observe that the absence of either loss term leads to noticeable speech quality (MOSNet score) drop. We further test the model's sensitivity to these two loss weights and we are able to empirically conclude that our framework is not sensitive to the selection of these two hyper-parameters. When $\lambda_{SC}$ increases from 0.002 to 0.1 (by 50 times), SV accuracy only fluctuates by $\pm 0.45\%$ and MOSNet score only fluctuates by $\pm 0.01$. When $\lambda_{VQ}$ increases from 0.01 to 2.0 (by 200 times), SV accuracy only fluctuates by $\pm 0.05\%$ and MOSNet score only fluctuates by $\pm 0.02$.

In a nutshell, both structural constraint and VQ diversity loss are necessary, and our model is not sensitive to the selection of the corresponding loss weights in a reasonably large range.

