# OpenReview forum: "Retriever: Learning Content-Style Representation as a Token-Level Bipartite Graph"
_ICLR.cc/2022/Conference — ICLR 2022 Poster_

### Official Review · Reviewer_c2Ut · 2021-11-01

**Correctness:** 3
**Technical Novelty And Significance:** 3
**Empirical Novelty And Significance:** Not applicable
**Recommendation:** 6
**Confidence:** 3

**Main Review:**

Strengths:

S1: This paper addresses an important problem -- content and style decomposition representation learning and proposes an unsupervised framework to learn content-style representations.

S2: The proposed model performs relatively well on both speech and image domains.

Weaknesses:

W1: It is difficult to follow the paper some notations lack details. For example, what are the prototypes and link keys in Figure 3 and how to obtain them? For speech and image domains, how to obtain the input of the model?

W2: Lack of ablation study on hyper-parameters in Eq. (1). What are the roles of the three items in Eq(1)?

W3: It is unclear how the model avoids or mitigates information leaking. It seems that a perfect model should avoid the style leak while achieving perfect reconstruction (shown in Figure 2 (c)).


**Summary Of The Paper:**

This paper proposes an unsupervised framework to learn content and style representations of structured data, such as texts, images and speeches. The task is interesting to me and decomposition representation learning is important. In addition, a bipartite graph is proposed to model the link between content tokens and style tokens, making the model more interpretable. The proposed model performs relatively well on both speech and image domains.

**Summary Of The Review:**

This paper addresses an important problem and proposed a framework to learn content and style representations. The experiments show that the performance of the proposed model is relatively well. But some details are missing, so it is difficult to follow the paper and it is unclear the roles of the loss functions and how the model avoids information leaking.

---

> ### Author Response · Authors · 2021-11-23
> **Response to Reviewer c2Ut**
>
> The authors would like to thank the reviewer for the positive comments on our work and the constructive suggestions.
>
> 1. Clarification of some notations (W1).
>
> We understand your concern and would like to clarify some notions here for better understanding.
> -	Prototypes, linking keys:
>
> Both the prototypes and linking keys are learnable network components. They are randomly initialized and optimized via back-propagation through style encoder and link attention decoder, respectively.
>
> -	The input to Retriever:
>
> It is discussed in main paper section 5.1.1 and 5.2.1. More details are shown in Appendix E1 and F1. In speech domain, we follow the previous work to use the CPC feature. The input to Retriever is the output of a depth-wise conv layer (kernel size 15) applied on CPC feature. In image domain, we down-sample image by 4 times using a stack of convolutions (detailed structure is shown in Appendix Table 5) to get the input tokens.
>
> 2. Ablation study on hyper-parameters in Equation 1 (W2).
>
> Thank you for pointing this out. We have added the ablation study both in image domain (appendix E.3) and speech domain (appendix F.7). First let us introduce the different roles of the three loss items as follows:	(i) reconstruction loss, the main loss to restrict that content-style representations capture the complete information of the structured input data. (ii) VQ diversity loss: a regularization on VQ codes to encourage all the codes to be equally used. (iii) structure constraint: a regularization on content for interpretable representation based on the a priori that interpretable representation usually requires spatial-temporal continuity.
>
> For the ablation results, we find that it is crucial to have both VQ diversity loss and structure constraint. However, the performance is quite stable at a relatively large range for most of the hyper-parameters. For example, In Appendix F.7, when $\lambda_{SC}$ changes from 0.002 to 0.1 (increased by 50 times), SV accuracy only fluctuates by $\pm 0.45%$, and MOSNet score only fluctuates by $\pm 0.01$. When $\lambda_{VQ}$ changes from 0.01 to 2.0 (increased by 200 times), SV accuracy only fluctuates by $\pm 0.05%$ and MOSNet score only fluctuates by $\pm 0.02$. In Appendix E.3, when $\lambda_{VQ}$ changes from 0.01 to 0.3 (increased by 30 times), landmark regression error fluctuates by only $\pm 0.72$.
>
>
> 3. information leaking (W3).
>
> We would like to provide clarification on Figure 2 to address your concern. Note that Figure 2 not only indicates what a good disentanglement is, but also indicates a clear strategy on how to achieve good disentanglement for our model. As shown in Figure 2, the three possible states of our model are style info leak (a&b), perfect disentanglement (c), and content info loss (d). The state only depends on the bottleneck size (in bule color). Therefore, the strategy is tuning the information bottleneck size and monitoring two metrics: reconstruction quality and style transfer similarity. More specifically, if reconstruction is poor, that means content information is lost, then the bottleneck should be tuned wider. Otherwise, if style transfer similarity is poor, that means style information is leaked into content. Then the bottleneck should be tuned narrower. In our main paper Table 3, the comparison among different bottleneck sizes demonstrates this tuning strategy. This is the way how we find an appropriate bottleneck and avoid information leak.

---

### Official Review · Reviewer_1d43 · 2021-11-02

**Correctness:** 3
**Technical Novelty And Significance:** 3
**Empirical Novelty And Significance:** 2
**Recommendation:** 6
**Confidence:** 3

**Main Review:**

*Strengths*
- S1. The proposed method is multimodal and can serve multiple modalities with a broad definition of style and content. The paper provides results on speech and vision data.
- S2. The proposed method is unsupervised, so it does not need annotation on style/content for the training data. This is very useful, as the boundaries on style and content are not straightforward and challenging to annotate.
- S3. The paper is clear and easy to read.
- S4. Experiments are technically sounds.

*Weaknesses*
- W1. The definition of style as "the information that is not affected by the permutation of tokens" is a bit counterintuitive — no justification or examples of why the paper chooses this, and no other definitions are provided.
- W2. The definition used in the paper does not correspond to the traditional definition of style and content in computer vision, which has been (and still is) a widely studied problem. This creates some confusion, especially in the image domain section.
- W3. After multiple examples referring to text as a potential target modality for this framework, no experiments are conducted on NLP datasets.
- W4. Additionally, for the image domain experiments, it would have been interesting to see results on art datasets, a field in which the problem of style and content disentanglement has been severally studied.
- W5. In the image domain experiments, there is no comparison to other methods other than for the proxy task of landmark regression. As visual results on style transfer are provided, it is expected to compare against other methods to see the potential of the proposed model against current technology.

**Summary Of The Paper:**

This paper proposes a framework for learning disentangled representations of content and style in an unsupervised way. The paper presents a slightly uncommon definition of content and style, which serves as a foundation for the proposed methodology and experiments. Experiments are conducted on speech and image datasets.

**Summary Of The Review:**

The idea of the paper is interesting and potentially useful for the community. However, the experimental results seem limited for now.

---

> ### Author Response · Authors · 2021-11-23
> **Response to Reviewer 1d43**
>
> The authors would like to thank the reviewer for the careful review and the inspiring questions.
>
> 1. Uncommon definition of content and style (W1&W2)
>
> We find that W1 and W2 raised by the reviewer are both related to the uncommon definition of content and style. We understand your concern. Below are some intuitions about our definition:
>
> In image domain, one common definition of content and style is shape and appearance. Our definition is partially aligned with it. However, researchers are having different definitions for appearance. In artistic style transfer, the appearance (style) is focused on low-level features [a], such as the overall color and texture, while the appearance in some other work is associated with semantically meaningful contents [b, c], e.g., the appearance of the eye or mouth. Our definition is more aligned with the latter category.
>
> In speech domain, AdaIN-VC [d] defines that style is the global statistical information and content is what is left after instance normalization (IN). Similar to the artistic style definition in the image domain, the style defined by IN operation is statistics of low-level features (e.g., the pitch and timbre of a speaker), while our style is related to semantically meaningful contents (e.g., how a speaker pronounces different phonemes.)
>
> [a] Arbitrary Style Transfer in Real-time with Adaptive Instance Normalization. ICCV 2017.
>
> [b] Deforming autoencoders: Unsupervised disentangling of shape and appearance. ECCV 2018.
>
> [c] Unsupervised Part-Based Disentangling of Object Shape and Appearance. CVPR 2019.
>
> [d] One-shot voice conversion by separating speaker and content representations with instance normalization. Interspeech, 2019.
>
>
> 2. No experiments on NLP datasets. (W3)
>
> Thanks for this comment. We believe that text, as a structured data, is a possible application domain of our framework. We plan to work on it in the future. For the time being, we choose to demonstrate the generalization capability of Retriever through the experiments in the speech and image domains, which, to be frankly, are more familiar to the co-authors of this paper. If the reviewer thinks that the current writing will mislead readers into thinking that we have tried Retriever on NLP tasks, we can reduce the description of the text modality in the final version.
>
> 3. Results on art datasets. (W4)
>
> We provide some results of the style transfer using the CelebA-wild and WikiArt dataset in Figure. 20 of Appendix E.4. Note that the proposed Retriever framework is designed for content-style decomposition. Although style transfer is one of its important downstream tasks, there is an important difference here. A content-style decomposition framework assumes that the content provider and the style provider are from the same domain, e.g., they are both human face images. It does not accept arbitrary statistics-based style as AdaIN does. Therefore, we cannot transfer the style of a human face by directly using an artistic painting. In order to demonstrate the results of artistic style transfer, we first use AdaIN to transfer the artistic style to a “prototypical face,” and then use this face as the style provider.
>
> The results shown in Appendix E.4 are zero-shot results, i.e., the network is not re-trained for these unseen data. Interestingly, Retriever manages to capture the tone and the appearance of facial organs of the style provider, and successfully transfers them to the content provider. Meanwhile, the shape (the layout and orientation of facial organs) of the content provider remains. This is exactly what we can expect from Retriever. We notice that our transferred images are not sharp, and some details are lost. This is because the original model is trained with a tight bottleneck. If the network is retrained with a relaxed bottleneck, we can expect a higher reconstruction quality.
>
> 4. Compare with other style transfer methods. (W5)
>
> We appreciate this comment. We have included visual comparison with other style transfer methods on DeepFashion dataset in Figure. 21 and Figure. 22 of Appendix E.4. The results show that Retriever achieves more natural style transfer at both part level and global level.

---

### Official Review · Reviewer_WCnU · 2021-11-03

**Correctness:** 4
**Technical Novelty And Significance:** 3
**Empirical Novelty And Significance:** 4
**Recommendation:** 8
**Confidence:** 4

**Main Review:**

Interesting idea of content and style disentanglement; validated on both image and audio domain.
# Strength:
1.	The paper proposes a new way to define style (for a given dataset) using permutation invariant (P.I) network. This is a good attempt to have a common definition of style.
2.	The network architecture is novel although using existing components such as VQ network and attention network. The idea of disentangling and link attention are what make the paper interesting.
3.	The model is domain-agonistic – demonstrated to work well in audio and image domain
4.	The paper demonstrates various downstream task in each domain. The part style transfer is impressive although there are some artifacts. Nice that the authors provided audio sample to play in supplementary file.

# Weakness/Suggestion/Queries

The paper proposes to separate out style representations. But it lacks visualization of how actually each individual style visually look like. It may be that these styles are actually relates to attribute of the dataset which for some datasets (e.g. DeepFashion, CUB birds) already defined. It would be interesting to cluster these style features and see the discovered styles from a given dataset.

Overall, there seems to a lot of hyperparameters to tune and may be that one must have good understanding of dataset for proper hyperparameter settings. Details as follows:

1.	How is $r_{\theta}$ implemented? It seems this is the linking attention network. Could you please include some discussion with guiding equations at least in the supmat.
2.	Content Encoder: It is stated  the spatially adjacent tokens are forced to share the same VQ code. If so, how does this work when there are two different objects (parts) in the adjacent tokens (adjacent patches in images)?
3.	$\mathcal{Z}_o \in \mathbb{R}^{m \times d_s} $: What is a suitable value for m? Does the value of m relates to the number of styles the network can represent?
4. The total loss is sum of three loss terms with $\lambda$ weighting parameters. How sensitive is the model (training/convergence) to the hyper-parameters $\lambda$ s? If the choice of these parameters slightly off, do the network perform still reasonably?
Any training strategy used to deal with discrepancies brought by these parameters? How do you train the network in different stages e.g., adding more loss terms in subsequent stage?

5.	How do you come up with G=2 and V=100 VQ is sufficient to capture all content information for audio dataset? Any heuristics for this, or perform rigorous parameter search?
6.	Do you do manual observation to associate content codes with parts in image domain? It is stated that V is dependent on image datasets. Does this mean that you set V to the number of parts you want to segment out from images?
7. It is not clear how the authors generate: (a) Codebooks for content (b) Prototype vectors for style?



**Summary Of The Paper:**

The paper proposes a novel idea of disentangling the content and style of structured data by treating style as permutation invariant information. It adopts VQ network for content encoding, and Cross-Attention for Style and Linking Attention at decoder. It is shown to be domain agonistic - worked well in image and audio domain.

**Summary Of The Review:**

The idea is interesting and seems novel, and the method is domain agonistic that may be applied to other domains. Experiement conducted are convincing, I am leaning to accept the paper, provide the minor concerns raised are resolved.

---

> ### Author Response · Authors · 2021-11-23
> **Response to Reviewer WCnU**
>
> The authors would like to thank the reviewer for acknowledging the new way of content-style definition and the novel network architecture to implement it. We appreciate your careful review and constructive suggestions.
>
> 1. Visualization of discovered styles (in response to the general suggestion)
>
> Thanks for this constructive suggestion. We visualize the style in both speech and image domain and the results are shown in Appendix C. Indeed, the clusters of style feature are related to the attributes of datasets in both image domain and speech domain. For example, in speech domain, the style vectors correspond to speaker id as shown in Appendix Figure 11. In image domain, some style vectors are related to “clothes color” and some others are related to gender of the model, as shown in Appendix Figure 12, 13.
>
> 2. Implementation details (in response to the detailed queries denoted by Q1-Q7)
>
> 2.1 How is $r_{\theta}$ implemented (Q1)
>
> Yes, $r_{\theta}$ is the major part of link attention. Specifically, it is the weight calculated from Q (query) and K (key) as in the standard attention mechanism. The equation is written as follows:
> $$ r_{\theta}(c_i, k_j) = Softmax(\frac{Q(c_i)\cdot K(k_j)}{\sqrt{d}}),$$
> Where $Q(c_i), K(k_j)$ are projection functions implemented by linear layers, $d$ is the dimension of the projected query and key vectors, and the softmax operation is done on index $j$,
>
> 2.2 Content Encoder (Q2)
>
> Please note that apart from the structural constraint loss term, our major loss term is reconstruction loss. When the adjacent tokens belong to different objects or parts that have significantly different values (e.g., pixel values in images), the reconstruction loss will force them to be quantized into different VQ codes.
>
> 2.3 Suitable value for $m$ (Q3)
>
> Yes, $m$ is related to the number of styles the network can represent. As shown in Appendix F.7, Figure 23, larger $m$ leads to better performance. Therefore, we just need to choose a relatively large $m$ to cover all P.I. information with consideration of the computation cost.
>
> 2.4 Loss weight sensitivity / training strategy (Q4)
>
> Our network does not need multiple-stage training, and most of our hyper-parameters are not sensitive. For example, In Appendix F.7, when $\lambda_{SC}$ changes from 0.002 to 0.1 (increased by 50 times), SV accuracy only fluctuates by $\pm 0.45%$, and MOSNet score only fluctuates by $\pm 0.01$. When $\lambda_{VQ}$ changes from 0.01 to 2.0 (increased by 200 times), SV accuracy only fluctuates by $\pm 0.05%$ and MOSNet score only fluctuates by $\pm 0.02$. In Appendix E.3, when $\lambda_{VQ}$ changes from 0.01 to 0.3 (increased by 30 times), landmark regression error fluctuates by only $\pm 0.72$.
>
> 2.5 Vector quantization: How to decide G=2 and V=100 in speech domain (Q5)
>
> In speech representation learning, it is a common practice to use multiple groups in VQ. For example, in [a], G is set to 2, so we follow this practice. It is believed that the two groups are encoding different levels of content, e.g., the phonemes and the detailed variations inside each phoneme.
> V is set to cover the number of phonemes (ideally also cover the possible variations for each phoneme). In English, usually 42 to 52 phonemes are defined (depending on different definitions), so we simply set V=100 and empirically find it to be a reasonable value.
>
> [a] Baevski et al. wav2vec 2.0: A framework for self-supervised learning of speech representations. NeurIPS, 2020.
>
> 2.6 Vector quantization: How to decide V in image domain? (Q6)
>
> Similar to the V selection in the speech domain, V is selected to cover the number of parts one wants to segment out from the images. In our experiments for image domain, we follow previous works to show the two choices of V in Table 4.
>
> 2.7 How to generate content codebooks and prototype vectors for styles (Q7)
>
> Both of them are learned. They are randomly initialized and then updated by the Adam optimizer. As discussed in appendix A, VQ output is a weighted sum of codewords in codebooks. Together with Gumbel-Softmax method, the gradient can be passed to VQ codebooks. For prototype vectors of style, their gradient can be back-propagated through style encoder.

---

### Decision · Program_Chairs · 2022-01-20

**Decision:**

Accept (Poster)

**Comment:**

This paper proposes a framework for learning disentangled representations of content and style in an unsupervised way, using a permutation invariant network. It adopts VQ network for content encoding, and Cross-Attention for Style and Linking Attention at decoder. It is shown to be domain agonistic, working well in image and audio domain. Experiments are conducted on speech and image datasets.

The paper is recommended as an accept (weak) to ICLR. The reviewers have given detailed feedback and suggestions -- please address them in the next revision of the paper.